# Distinct T-cell receptor (TCR) gene segment usage and MHC-restriction between foetal and adult thymus

**Jasmine Rowell[1], Ching-In Lau[1], Susan Ross[1], Diana C Yanez[1], Oscar A Peña[2], Benny Chain[3], Tessa Crompton[1]***

[1]UCL Great Ormond Street Institute of Child Health, London, United Kingdom; [2]School of Biochemistry, University of Bristol, Bristol, United Kingdom; [3]Division of Infection and Immunity, University College London, London, United Kingdom

## eLife Assessment

This **important** manuscript provides an extensive and **convincing** analysis of the foetal and adult TCR repertoire in the mouse thymus. A potential implication of the work is that the earliest appearing T cells during ontogeny may have properties that are fundamentally distinct from those appearing later in life. The study will be of interest to immunologists concerned with T cell development and TCR repertoires.

***For correspondence:**
t.crompton@ucl.ac.uk

**Competing interest:** The authors declare that no competing interests exist.

**Abstract** Here, we sequenced rearranged TCRβ and TCRα chain sequences in CD4$^+$CD8$^+$ double positive (DP), CD4$^+$CD8$^-$ single positive (SP4) and CD4$^-$CD8$^+$ (SP8) thymocyte populations from the foetus and young adult mouse. We found that life-stage had a greater impact on TCRβ and TCRα gene segment usage than cell-type. Foetal repertoires showed bias towards 3'TRAV and 5'TRAJ rearrangements in all populations, whereas adult repertoires used more 5'TRAV gene segments, suggesting that progressive TCRα rearrangements occur less frequently in foetal DP cells. When we synchronised young adult DP thymocyte differentiation by hydrocortisone treatment the new recovering DP thymocyte population showed more foetal-like 3'TRAV and 5'TRAJ gene segment usage. In foetus we identified less influence of MHC-restriction on α-chain and β-chain combinatorial VxJ usage and CDR1xCDR2 (V region) usage in SP compared to adult, indicating weaker impact of MHC-restriction on the foetal TCR repertoire. The foetal TCRβ repertoire was less diverse, less evenly distributed, with fewer non-template insertions, and all foetal populations contained more clonotypic expansions than adult. The differences between the foetal and adult thymus TCR repertoires are consistent with the foetal thymus producing αβT-cells with properties and functions that are distinct from adult T-cells: their repertoire is less governed by MHC-restriction, with preference for particular gene segment usage, less diverse with more clonotypic expansions, and more closely encoded by genomic sequence.

## Introduction

T-cells are essential for adaptive immunity to mount an immune response against a wide and unpredictable array of foreign pathogens. T-cells recognise pathogens through their T-cell receptor, and adaptive immunity relies upon a broad repertoire of T-cells with TCRs with different specificities to potentially unknown pathogens, so that should a new threat arise, the immune system can respond. The diverse repertoire of possible TCRs is generated from gene rearrangement of two independently generated chains, the β chain and α chain, which come together to form the TCR. This diverse pool

of αβTCR is generated during T-cell development in the thymus by the RAG-dependent joining of variable (V), joining (J), and diversity (D) gene segments at the β locus, and V and J gene segments at the α locus, with additional diversity coming from non-template nucleotides (junctional diversity; *Schatz and Swanson, 2011*).

The thymus is essential for both T-cell development and TCR repertoire selection, but undergoes changes in output and function across the life of a mouse (*Kondo et al., 2019*; *Montecino-Rodriguez and Dorshkind, 2023*). In this study, we compared the αβTCR repertoire generated in the foetal thymus to that of the young adult, to investigate how these life-stages influence V(D)J gene usage and TCR repertoire diversity and distribution.

The thymus and the parathyroid organs, emerge from the third pharyngeal pouch and cleft during mid gestation of the foetal mouse (*Gordon and Manley, 2011*). Haematopoietic progenitors migrate from the foetal liver through the blood stream to colonise the thymus from embryonic day (E) 10–12. The first CD4⁻CD8⁻ double negative (DN) thymocytes are already present at E11 and proliferate over the following days, producing a larger population by E14. CD4⁺CD8⁺double positive (DP) and CD8+ immature single positive (ISP) thymocytes are first detectable at ~E16, while mature CD4⁺CD8⁻CD3⁺ single positive (SP4) and CD4⁻CD8⁺CD3⁺ single positive (SP8) thymocytes can be detected at day E18 (*Solanki et al., 2018*; *Solanki et al., 2020*). After birth, the thymus continues to grow rapidly and reaches its peak size at 4–6 weeks of age (*Xiao et al., 2003*).

During this developmental progression, the *Tcrb* and *Tcra* loci are sequentially rearranged to produce T-cells with a diverse TCR repertoire. Successful rearrangement of *Tcrb* and pre-TCR expression are required for commitment to the αβT-cell lineage and differentiation from the CD25⁺DN (DN3) stage to DP cell, whereas cells that successfully rearrange the γδTCR commit to the γδT-cell lineage at the CD25⁺ DN stage. *Tcra* rearrangement and signalling through the αβTCR are essential for MHC-restriction and differentiation from DP to SP cell (positive selection) and then also for deletion of T-cells bearing self-reactive TCR (negative selection). Pre-TCR signalling is required for survival, expansion, and differentiation of cells that have completed TCRβ rearrangement, to produce a pool of cells which express a single functional TCR β-chain, in which *Tcra* rearrangement and αβTCR repertoire selection will occur (*Dutta et al., 2021*) and β-selection is believed to involve immune synapse formation and MHC-interactions (*Allam et al., 2021*). In general, allelic exclusion of the TCR β-chain locus is believed to prevent developing thymocytes from simultaneously rearranging and expressing two TCR β-chains, whereas in DP cells the TCR α-chain locus undergoes biallelic Vα-Jα rearrangement, with many possible rounds of rearrangement on each allele (*Carico and Krangel, 2015*; *Carico et al., 2017*; *Genolet et al., 2012*). Although the sequence of this developmental programme is the same between foetal and adult αβT-cell development, progression beyond the DN3 stage is less tightly linked to TCRβ expression and β-selection in the foetal thymus (*Hager-Theodorides et al., 2007*).

There are several other fundamental differences between adult and foetal thymus that may affect the generation and selection of the TCR repertoire. In contrast to adult αβT-cells which are produced in a thymus structure that is mature, during foetal T-cell development, the thymus is still forming: the structure of the medulla and differentiation of the medullary thymic epithelial cell (mTEC) population, which express tissue restricted antigens for tolerance induction, are dependent on the haematopoietic compartment, with different requirements in foetal and adult thymus (*Desanti et al., 2012*). There are also differences in the progenitor cells that seed the thymus at different life-stages (*Ramond et al., 2014*), which arise from either the foetal liver or postnatal bone-marrow, and enter the thymus through different locations (*Montecino-Rodriguez and Dorshkind, 2023*). Some foetal haematopoietic stem cells (HSC) have been shown to have increased capacity to proliferate and to differentiate preferentially into innate-like lymphocytes compared with adult cells (*Beaudin et al., 2016*). Additionally, CD8 T-cells derived from HSCs from the foetal liver are functionally distinct from adult CD8 T-cells (*Wang et al., 2016*).

The foetal TCR repertoire is known to be less diverse than the adult repertoire. TdT is absent from the foetal thymus and is first expressed a few days after birth, so foetal repertoires contain less non-template nucleotide additions (*Bogue et al., 1992*). This has been confirmed by next generation TCR sequencing from mouse and human foetal and adult repertoires, which showed increased diversity after birth that was attributed to the postnatal increase in non-template nucleotide additions (*Sethna et al., 2017*; *Britanova et al., 2016*; *Pogorelyy et al., 2017*). Additionally, a PCR approach has

indicated that the foetal TCRα repertoire uses a limited range of VJ rearrangements with enrichment of proximal VJ rearrangements (*Pasqual et al., 2002*).

In this study, we compared the TCR repertoire that is generated and selected in the foetal thymus with that from the young adult thymus. In addition to the reduction in non-template nucleotide additions seen in the embryonic repertoire, we identified biases in VJ usage and VxJ pairing of both TCRβ and TCRα repertoires. The foetal TCRβ repertoires were less diverse and less equally distributed with more clonal expansions than the young adult repertoires.

## Results

### Foetal thymocyte populations show clonotypic TCRβ expansion with a less diverse TCRβ repertoire than adult

To compare the TCRβ chain repertoire generated in the embryonic and young adult thymus, we FACS-sorted DP (CD4⁺CD8⁺), SP4 (CD4⁺CD8⁻CD3⁺), and SP8 (CD4⁻CD8⁺CD3⁺) populations from E18.5 (foetal) and 4-week-old (young adult) thymus and sequenced their rearranged TCR transcripts. After RNA extraction from purified populations, we used a TCR sequencing protocol and analysis pipeline which includes single-strand DNA ligation that tags each molecule of *Tcra* and *Tcrb* mRNA with a unique molecular identifier (UMI), allowing PCR bias to be corrected for in later analysis (*Oakes et al., 2017*; *Uddin et al., 2019*; *Thomas et al., 2013*). Firstly, to visualise the frequency distribution of the TCRβ repertoire in each population, we plotted the TCRβ abundance (number of copies of each sequence) against their proportion of the repertoire (*Figure 1A*). For each of the developmentally defined thymocyte populations, the foetal thymus showed an increase in the proportion of abundant clones compared to adult mice in DP, SP4, and SP8 populations (*Figure 1A*). The frequency distributions of TCRβ abundances were fitted to a discrete power law, where the power law exponent corresponds to the gradient on the log-log plots (*Figure 1A*). The power law exponent was significantly lower in each foetal population compared to its adult counterpart (*Figure 1B*), demonstrating a difference in distribution of abundance of TCRβ clones between foetal and adult, consistent with an increase in clonotypic expansion of abundant clones in the foetal TCRβ repertoire. To confirm this, we calculated the proportion of the TCRβ repertoire represented by the top 1% most abundant TCRβ clones in each population. In foetal DP and SP4 and SP8 populations, the proportion of the TCRβ repertoire represented by the top 1% most abundant TCRβ clones was more than twice that of the adult populations (*Figure 1C*), and the mean abundance of TCRβ sequences detected in the top 1% most abundant (expanded) clones was also significantly higher (*Figure 1D*), in the foetal SP4 population only, suggesting a difference in equality of distribution in the top 1% most abundant (expanded) clones in the DP and SP8 populations compared to SP4 (*Figure 1D*).

We therefore calculated standard indices of diversity, richness and distribution: Shannon entropy and the Gini index. Shannon entropy is derived from information theory and represents the complexity of the system, taking into account both the number of unique clonotypes and the frequencies of each clonotype, to generate a measure of richness and diversity (*Shannon and Weaver, 1949*). For each thymocyte population, the Shannon entropy was higher in the young adult TCRβ repertoire than in the foetal, demonstrating a richer more diverse repertoire in the adult thymocyte populations (*Figure 1E and F*). The Gini index is a measure of the equality (evenness) of the distribution, and is frequently used in economics to demonstrate the distribution of a population's wealth, where a lower index represents a more equal distribution (*Ceriani and Verme, 2012*). The Gini index of the distribution of TCRβ clonotypes was significantly lower in each adult population compared to its foetal counterpart, confirming a more equally distributed TCRβ repertoire in the young adult thymus (*Figure 1G and H*).

In parallel, we analysed the repertoire diversity, richness, and distribution of the TCRα repertoires (*Figure 1—figure supplement 1*). The fitted power law exponent of the frequency distribution of TCRα abundances was significantly lower in the foetal DP and SP4 populations compared to their adult counterparts (*Figure 1—figure supplement 1A and B*), whereas the proportion of the TCRα repertoire represented by the top 1% most abundant TCRα clones was higher in the foetal DP population than the adult DP population (*Figure 1—figure supplement 1C*). However, we detected no significant differences between foetal and adult populations in the abundance of TCRα sequences detected in the top 1% most frequent (expanded) clones (*Figure 1—figure supplement 1D*). The Shannon entropy was significantly lower in each foetal population compared to its adult counterparts

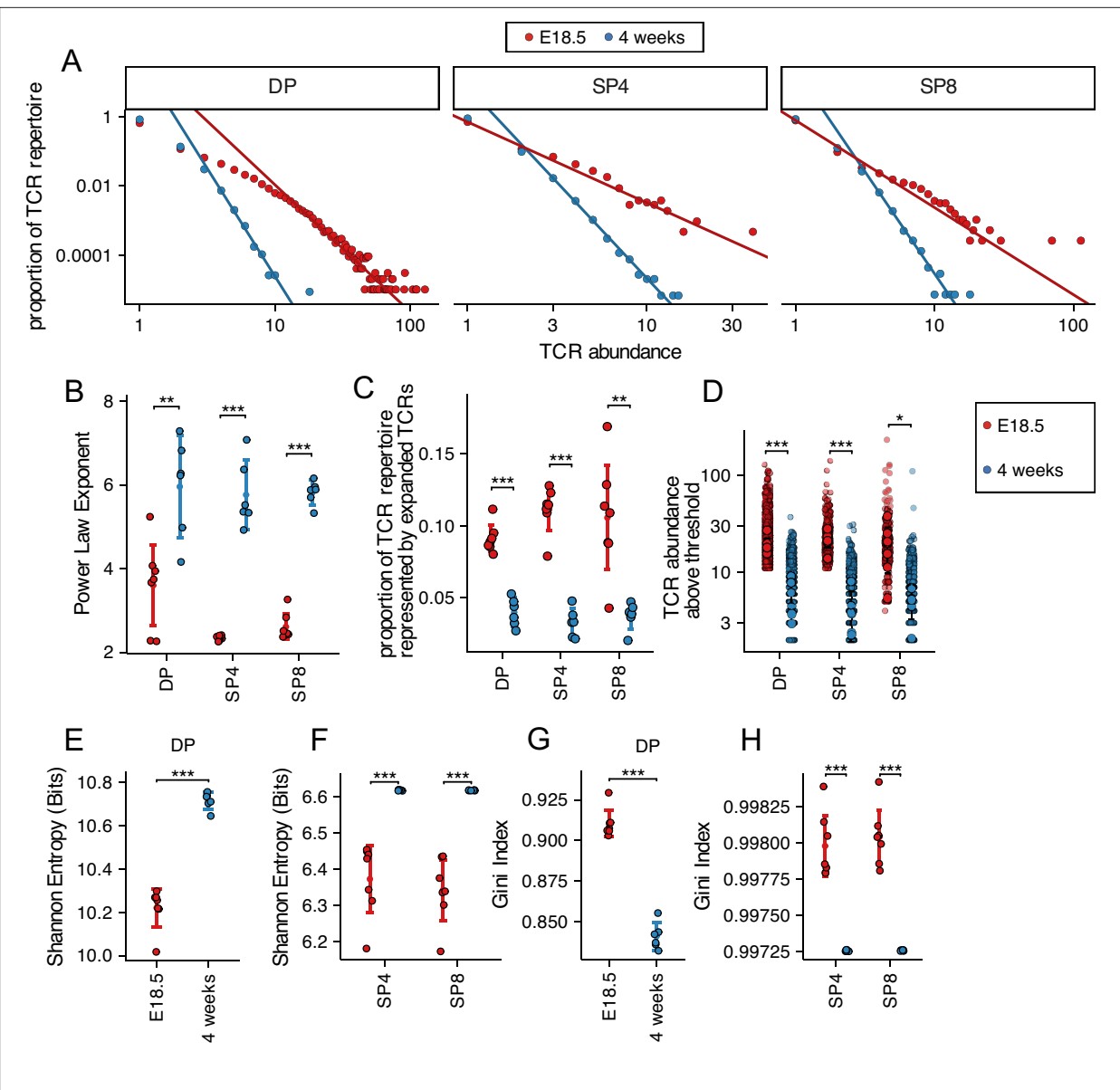

**Figure 1.** Decreased diversity and increased clonality of foetal compared to young adult TCRβ repertoires. TCR β-chain repertoires were sequenced from FACS-sorted CD4+CD8+ (DP), CD4+CD8-CD3+ (SP4), and CD4-CD8+CD3+ (SP8) thymocyte populations from E18.5 (red circles; n=7) and 4-week-old (blue circles; n=6) C57BL/6 thymus. (**A**) The frequency distribution of TCR β-chain abundance was fitted to a discrete power law ($f(k)=Ck^{-\alpha}$) by maximum likelihood (solid line). These logged plots show a representative individual mouse TCR repertoire frequency distribution for E18.5 and 4 weeks in DP, SP4, and SP8 populations. The x axis represents TCR abundance (size of clone), and the y axis represents the proportion of the repertoire. The negative of the power law exponent corresponds to the slope of the logged plot. (**B**) The power law exponents of the frequency distribution of TCR β-chain abundances for E18.5 and 4 weeks in DP, SP4, and SP8 populations, where each point represents an individual mouse/embryo. (**C**) The proportion of the total TCR β-chain repertoire accounted for by the expanded top 1% most abundant sequences (>99th percentile) for E18.5 and 4 weeks for DP, SP4, and SP8 populations, where each point represents an individual mouse/embryo. (**D**) The number of β-chain sequences detected above the given frequency threshold (top 1% most abundant sequences) is shown for E18.5 (red circles) and 4 weeks (blue circles) in DP, SP4, and SP8 populations. Each small translucent point represents the abundance of any β-chain sequence detected above the threshold for all mice/embryos at that life-stage, while each larger solid point represents the mean of β-chain sequence abundance for each individual mouse/embryo. (**E**) The Shannon entropies for the β-chain TCR repertoires of E18.5 and 4 weeks in DP populations, where each point represents an individual mouse/embryo. Each repertoire was subsampled to 50,000 TCRs 1000 times before calculating the Shannon entropy. (**F**) The Shannon entropies for the β-chain TCR repertoires of E18.5 and 4 weeks in SP4 and SP8 populations, where each point represents an individual mouse/embryo. Each repertoire was subsampled to 750 TCRs 1000 times before calculating the Shannon entropy. (**G**) The Gini indexes for the β-chain TCR repertoires of E18.5 and 4 weeks in DP populations, where each point represents an individual mouse/embryo. Each repertoire was subsampled to 50,000 TCRs 1000 times before calculating the Gini index. (**H**) The Gini indexes for the β-chain TCR repertoires of E18.5 and 4 weeks in SP4 and SP8 populations, where each point represents an individual mouse/embryo.

*Figure 1 continued on next page*

Figure 1 continued

Each repertoire was subsampled to 750 TCRs 1000 times before calculating the Gini index. Dotplots show mean ±c.i and statistical comparisons were carried out by unpaired Student's t-test or Welch's t-test as appropriate: *** p<0.001; ** p<0.01; * p<0.05.

The online version of this article includes the following figure supplement(s) for figure 1:

**Figure supplement 1.** Distribution and diversity of embryonic and young adult TCRα repertoire.

(*Figure 1—figure supplement 1E and F*), but the Gini index was significantly higher in each foetal population (*Figure 1—figure supplement 1G and H*), confirming that foetal TCRα repertoires were less rich, diverse and evenly distributed than young adult TCRα repertoires.

## Summary

Foetal DP, SP4 and SP8 TCRβ and TCRα repertoires are less evenly distributed and less diverse than their adult counterparts and foetal TCRβ repertoires contain more clonal expansions.

## Foetal and young adult TCRβ and TCRα repertoires use distinct TRAV, TRAJ, TRBV, and TRBJ gene segments

We next tested if the foetal and young adult repertoires showed differences in usage of TCR V and J gene segments. We found many differences between foetus and adult in proportional V and J segment usage within unique TCRβ and TCRα sequences for DP (*Figure 2—figure supplement 1*), SP4 (*Figure 2—figure supplement 2*), and SP8 (*Figure 2—figure supplement 3*) populations. Preferential gene segment usage according to life-stage appeared to correlate with the chromosomal location of the gene segment, particularly for TRAV and TRBV usage within the DP population, where the adult repertoire favoured the 5' gene segments, and the foetus the 3' gene segments. To visualise these differences, we plotted heatmaps of mean proportional gene segment usage for unique TCR sequences, clustering each life-stage and thymocyte population, but showing each gene segment in chromosomal order (*Figure 2A–D*). The samples clustered by life-stage first before cell-type (E18.5 populations clustered together) for both TCRα and TCRβ, suggesting that VJ gene usage is altered in the foetus compared to young adult. Both TRAV and TRAJ showed clear preference for chromosomal location, with foetal populations favouring 3' TRAV segments and 5' TRAJ segments, whereas adult populations showed enrichment of 5' TRAV and 3' TRAJ (*Figure 2A and B*). To test if this bias in gene usage was present in the unselected TCRα repertoire, we separated the DP population into cells that have not yet entered positive selection (CD3$^{-/lo}$DP/CD69$^-$DP) and cells that have entered TCR repertoire selection (CD3$^{+/hi}$DP/CD69$^+$DP). Both foetal DP populations clustered together and showed similar bias towards 3' TRAV and 5' TRAJ, suggesting that the bias was independent of positive selection. In the case of the adult DP populations, however, the more immature DP population clustered separately from the other young adult populations and showed weaker enrichment 3'TRAJ than the populations that had undergone or were undergoing positive selection (CD3$^{+/hi}$DP, SP4 and SP8). This suggests that in the adult thymus, positive selection changes proportional TRAJ gene usage, whereas in the foetal thymus an inherent bias in VJα gene usage persists after positive selection.

Heatmaps for TRBV also broadly showed chromosomal location-bias for gene usage at the different life-stages, with foetal populations preferentially using 3'TRBV, although the pattern appeared less pronounced (*Figure 2C and D*). Comparison of TRBJ cluster usage between adult and foetus showed that foetal SP TCRβ repertoires used proportionally more TRBJ gene segments from the TRBJ1 gene cluster than their adult counterparts, whereas adult SP repertoires used proportionally more TRBJ gene segments from the TRBJ2 cluster suggesting that positive selection influences TRBJ cluster usage (*Figure 2—figure supplement 4* and *Figure 2—figure supplements 1–3*). Therefore, to evaluate further the differences in TCRβ gene segment usage between foetal and adult repertoires, we carried out Principal Component Analysis (PCA) of the β-chain variable and joining gene counts of the unique TCRβ chains for each population (*Figure 2E–J*). PCA separated the repertoires for DP, SP4 and SP8 by life-stage on PC1, confirming differences in TRBV and TRBJ usage between life-stage for each population.

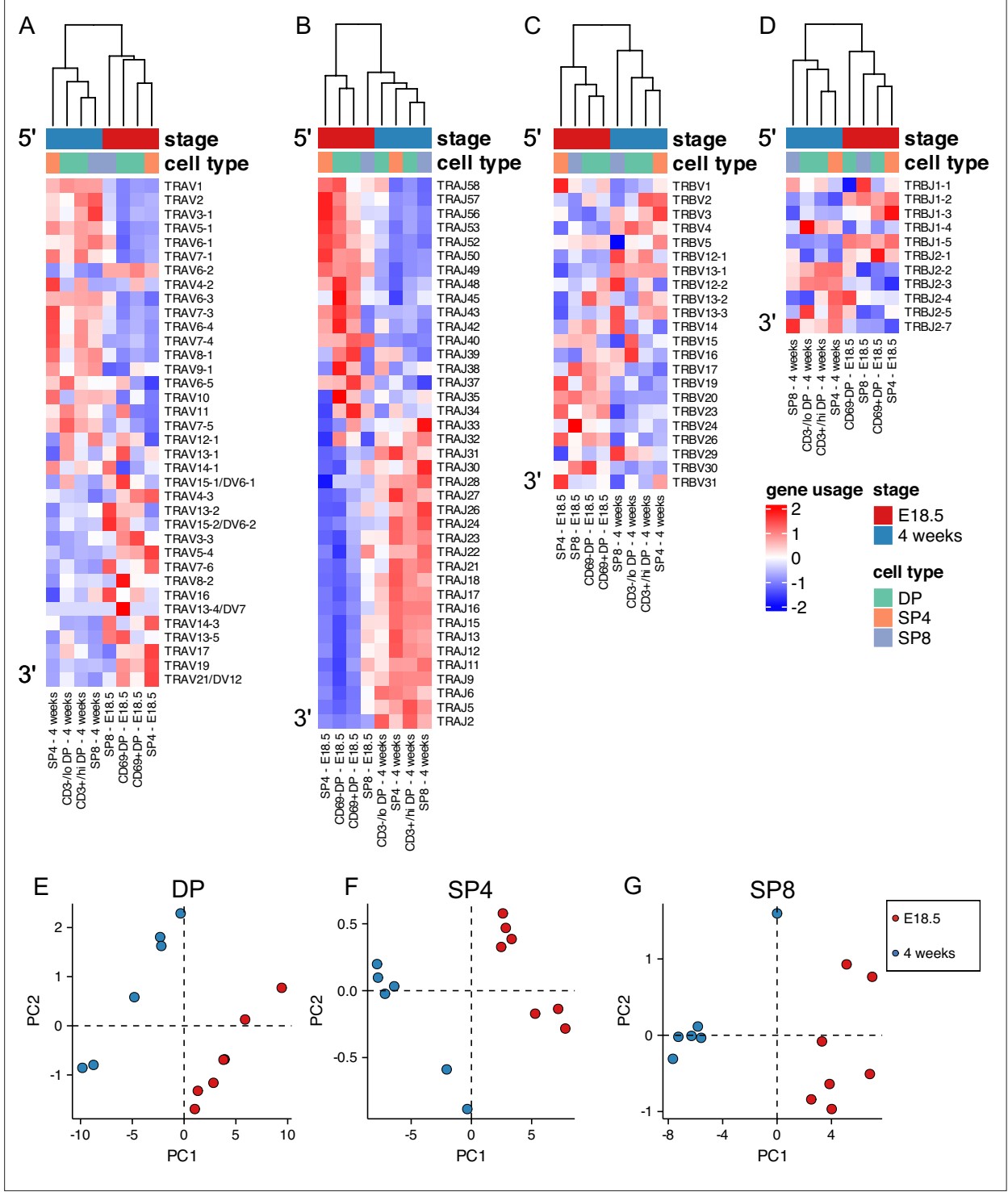

**Figure 2.** Foetal and young adult TCR repertoires favour different gene segments. TCR α-chain (**A, B**) and β-chain (**C–G**) repertoires were sequenced from FACS-sorted CD4$^+$CD8$^-$CD3$^+$ (SP4) and CD4$^-$CD8$^+$CD3$^+$ (SP8) CD4$^+$CD8$^+$CD69$^-$ (CD69-DP, foetal) CD4$^+$CD8$^+$CD69$^+$ (CD69$^+$DP foetal) CD4$^+$CD8$^+$CD3$^{-/lo}$ (CD3$^{-/lo}$DP, adult), CD4$^+$CD8$^+$CD3$^{+/hi}$ (CD3$^{+/hi}$DP adult) thymocyte populations from E18.5 (red; n=7) and 4 week-old (blue; n=6) C57BL/6 thymus. DP, SP4 and SP8 populations are coloured in green, orange and purple respectively. (**A–D**) Heatmaps of proportional α-chain variable (V) (**A**), α-chain joining (J) (**B**), β-chain V (**C**) and β-chain J (**D**) gene usage of unique TCRs for E18.5 and 4 weeks in DP, SP4 and SP8 populations. Each column represents a mean of 7 embryos or 6 mice and was clustered using Euclidian distance. Genes are shown in chromosomal order (5' to 3') from top to bottom. (**E–G**) PCA biplot of β-chain V and J gene counts of unique TCRs for E18.5 and 4 weeks in DP (**E**), SP4 (**F**) and SP8 (**G**) populations.

The online version of this article includes the following figure supplement(s) for figure 2:

*Figure 2 continued on next page*

**Figure supplement 1.** Foetal and young adult DP populations have proportionally different TRAV, TRAJ, TRBV and TRBJ gene usage.

**Figure supplement 4.** Proportional TRBJ gene segment usage from TRBJ1 and TRBJ2 gene clusters.

### Summary

Foetal and adult DP SP4 and SP8 thymocyte populations use distinct TCRβ and TCRα gene segments, with foetal populations showing bias towards 3' TRAV, 5' TRAJ, 3'TRBV and the TRBJ1 gene cluster.

## Foetal and adult TCRβ and TCRα repertoires use distinct combinations of V and J segments

Given this difference in TRBV and TRBJ gene usage between foetal and adult, we next tested if TCRβ foetal and young adult thymocyte repertoires use different combinations of V and J gene segments. We carried out PCA using the counts of each VxJ combination from unique TCRβ sequences in foetal

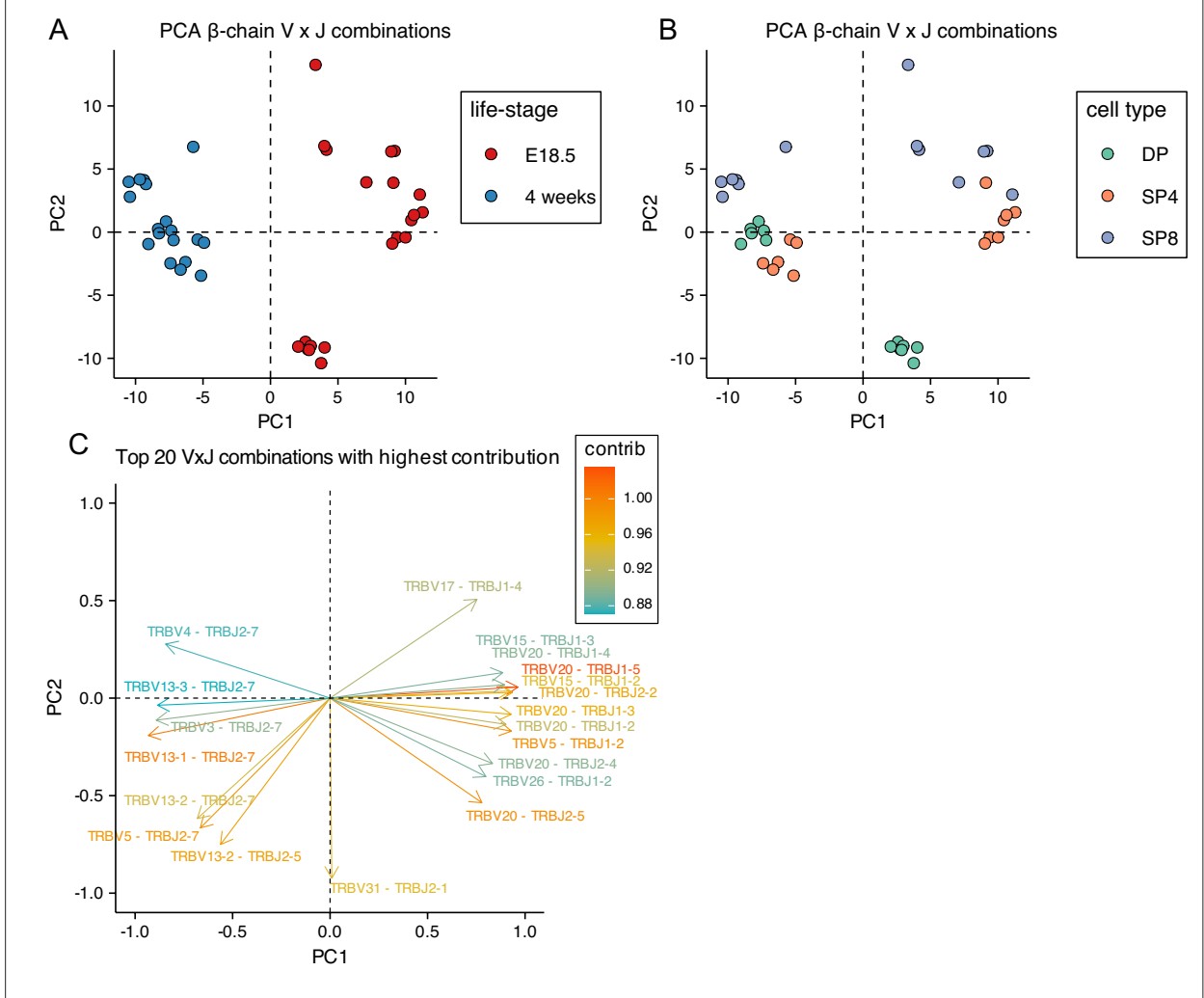

**Figure 3.** Principal Component Analysis (PCA) of Variable x Joining combinations in embryonic and young adult TCRβrepertoires cluster by life-stage before cell-type. TCR β-chain repertoires were sequenced from FACS-sorted CD4+CD8+ (DP), CD4+CD8-CD3+ (SP4) and CD4-CD8+CD3+ (SP8) thymocyte populations from E18.5 (n=7) and 4 week-old (n=6) C57BL/6 thymus. (**A, B**) PCA biplot of β-chain VxJ gene counts of unique TCRβ sequences for E18.5 and 4 weeks in DP, SP4 and SP8 populations. (**A**) E18.5 and 4 weeks are coloured in red and blue circles respectively. (**B**) DP, SP4 and SP8 populations are coloured in green, orange and purple respectively. (**C**) Top 20 highest β-chain VxJ gene combinations that contribute to PC1 and PC2 of the PCA.

and adult DP, SP4 and SP8 populations (*Figure 3A–C*). PCA separated by life-stage on PC1, with adult repertoires clustering tightly together (*Figure 3A*). When we compared by thymocyte population, each adult population clustered together, with adult SP4 separating from adult SP8 on PC2, and DP cells scoring in between, suggesting that PC2 corresponds to MHC-restriction of the adult populations (*Figure 3B*). In contrast, the foetal populations were more dispersed, and did not segregate fully by cell type on PC2 (*Figure 3B*). When we examined which VxJ combinations contributed most to PC1, we found strong correlation to the proportional TRBV and TRBJ usage detected: TRBV13-1 and TRBJ2-7 both showed adult bias in the analysis of individual gene usage, and in combination also contributed strongly to the negative score on PC1; whereas TRBV20 and TRBJ1-3 both showed foetal bias in the analysis of individual gene usage and contributed strongly to the positive (foetal) score on PC1, reflecting chromosomal gene segment location (*Figure 2D*, *Figure 2—figure supplements 1–3*).

We next compared usage of all possible VxJ combinations between the adult and foetal thymocyte populations for both TCRβ and TCRα repertoires, identifying all combinations that showed a significant change in proportional usage in foetus compared to young adult (*Figure 4A–C*). The DP foetal population showed many increases (86, shown in red) and decreases (72, shown in blue) in proportional usage of VxJ combinations for TCRβ repertoire compared to young adult, which after selection were reduced in the SP4 and SP8 populations, to 37 increases and 34 decreases in SP4, and 26 increases and 25 decreases in SP8. In the case of the TCRα chain, the foetal DP repertoire favoured VxJ combinations from the 3' location of TRAV and 5' location of TRAJ, and this bias persisted but was less pronounced in the SP populations (*Figure 4B and C*). Concomitantly, the DP foetal population showed clear proportional decrease in usage of combinations of 5' TRAV and 3' TRAJ in comparison to young adult.

## Summary
Foetal and adult DP, SP4 and SP8 thymocyte populations show bias in proportional combinatorial VxJ usage for both TCRα and TCRβ repertoires, with foetal TCRα repertoires enriched for proximal VJ rearrangements. PCA indicates that TCRβ combinatorial VXJ usage is less influenced by cell type (MHC-restriction) in foetal compared to adult repertoires.

## Shorter CDR3 length and reduced non-template insertion length in foetal TCRβ repertoires

We next tested if the percentage of non-productive TCRβ and TCRα rearrangements was different in foetal and young adult populations. Out of frame (non-productive) rearrangements generated during TCR gene rearrangement are eliminated by nonsense-mediated decay during T-cell development (*Weischenfeldt et al., 2008*), although some non-productive TCR transcripts persist (*Mahowald et al., 2011*; *Britanova et al., 2016*; *Camaglia et al., 2023*). The proportion of non-productive rearrangements was higher in each foetal population than adult (*Figure 5A* and *Figure 5—figure supplement 1A*).

Furthermore, we examined non-template additions/deletions to V and J gene segments, which in the TCRβ chain include TRBD nucleotides. TRBD segments are difficult to identify as they are highly homologous and short. As previously reported (*Sethna et al., 2017*), the mean TCRβ non-template nucleotide insertion length was lower in the foetal DP, SP4 and SP8 populations compared to adult. To confirm this we then investigated the length of the complementary determining region 3 (CDR3) of the TCRβ chain as it is the most hypervariable region and primary site of antigen recognition. The mean CDR3 length was shorter in all three populations in foetal TCRβ repertoires compared to young adult, confirming previous studies (*Sethna et al., 2017*, *Figure 5B and C*).

Likewise, when we compared amino acid CDR3 sequences for the TCRα repertoires, we found lower mean length of non-template insertions (*Figure 5—figure supplement 1B*) and mean α-chain CDR3 length in the foetal thymocyte populations compared to their adult counterparts (*Figure 5—figure supplement 1C*).

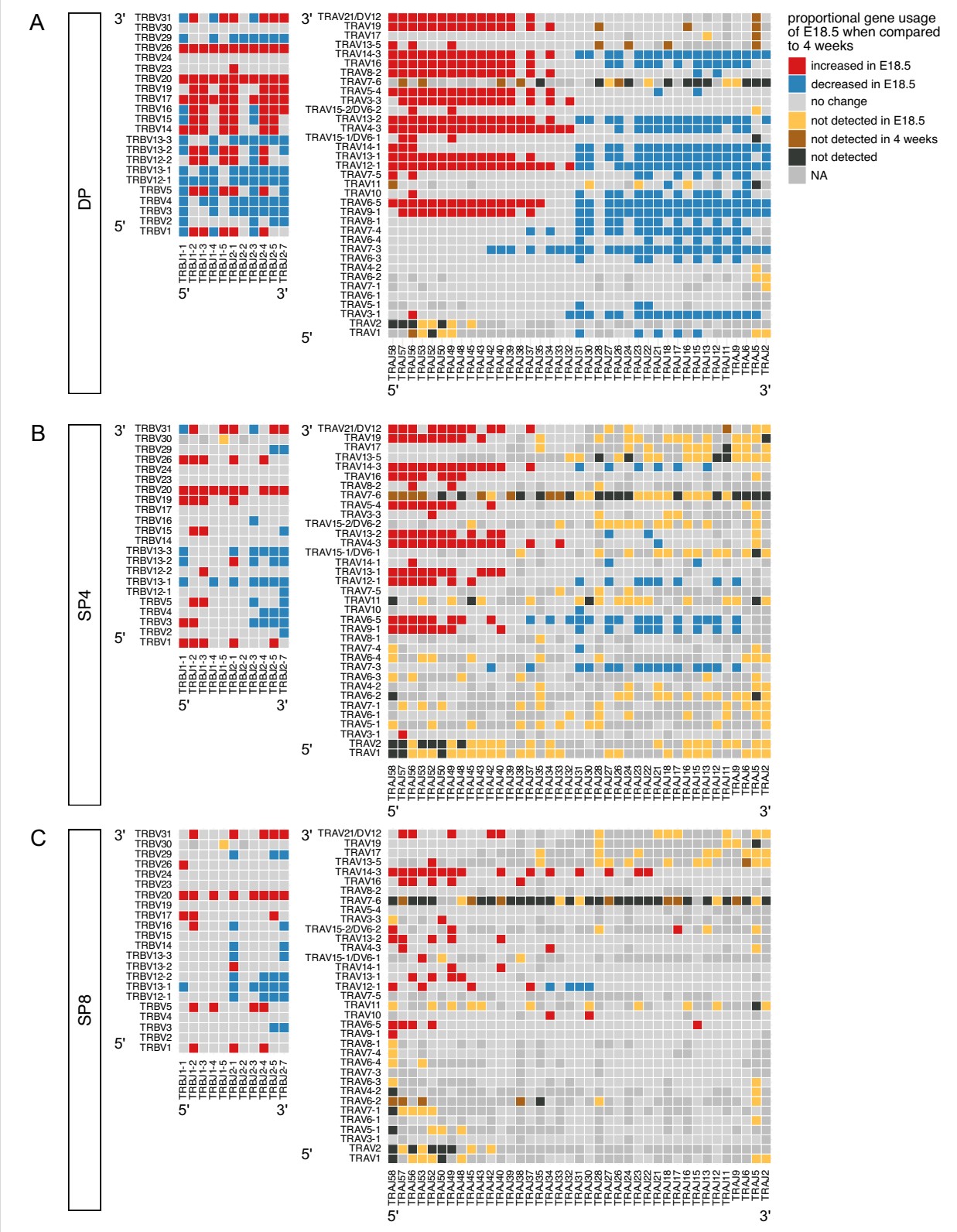

**Figure 4.** Proportional bias in VxJ combinations in foetal and young adult repertoires by cell-type. TCR β-chain and α-chain repertoires were sequenced from FACS-sorted CD4⁺CD8⁺ (DP), CD4⁺CD8⁻CD3⁺ (SP4) and CD4⁻CD8⁺CD3⁺ (SP8) thymocyte populations from E18.5 (red; n=7) and 4 week-old (blue; n=6) C57BL/6 thymus. (**A–C**) Plots show β-chain (left column) and α-chain (right column) proportional VxJ gene usage of total TCRs for E18.5 compared to 4 weeks in DP (**A**), SP4 (**B**) and SP8 (**C**) populations. Red tiles signify increased usage of VxJ combinations in E18.5 ($P < 0.05$), while blue

*Figure 4 continued on next page*

*Figure 4 continued*

tiles signify decreased usage in E18.5 (*P* < 0.05) compared to 4 weeks. Light grey tiles signify no change in gene usage (*P* > 0.05), yellow tiles signify VxJ combinations not detected in E18.5, brown tiles signify VxJ combinations not detected in 4 weeks and black tiles signify VxJ combinations not detected in both E18.5 and 4 weeks. Only combinations that were detected in at least 3 mice/embryos per group were compared. Dark grey tiles signify combinations that were not compared. Statistical comparisons were carried out by unpaired Student's t-test followed by FDR-adjustment (5%, Benjamini-Hochberg procedure) of *p* values.

## Sharing of CDR3 sequences differs between adult and foetal TCRβ and TCRα repertoires

In order to investigate sharing between the TCRβ CDR3 repertoires within the same cell population from different embryos/mice of the same life-stage, we used the Jaccard Index of Similarity. This index measures the size of the intersection divided by the size of the union of finite sample sets (*Jaccard, 1912*). We calculated the Jaccard Index of Similarity for each TCRβ CDR3 repertoire from each thymocyte population, comparing different mice/embryos to determine how similar each foetal repertoire is to the repertoire of the same thymocyte population from another foetus, and likewise for young adults. For each population, the Jaccard index was significantly higher in the foetal samples compared

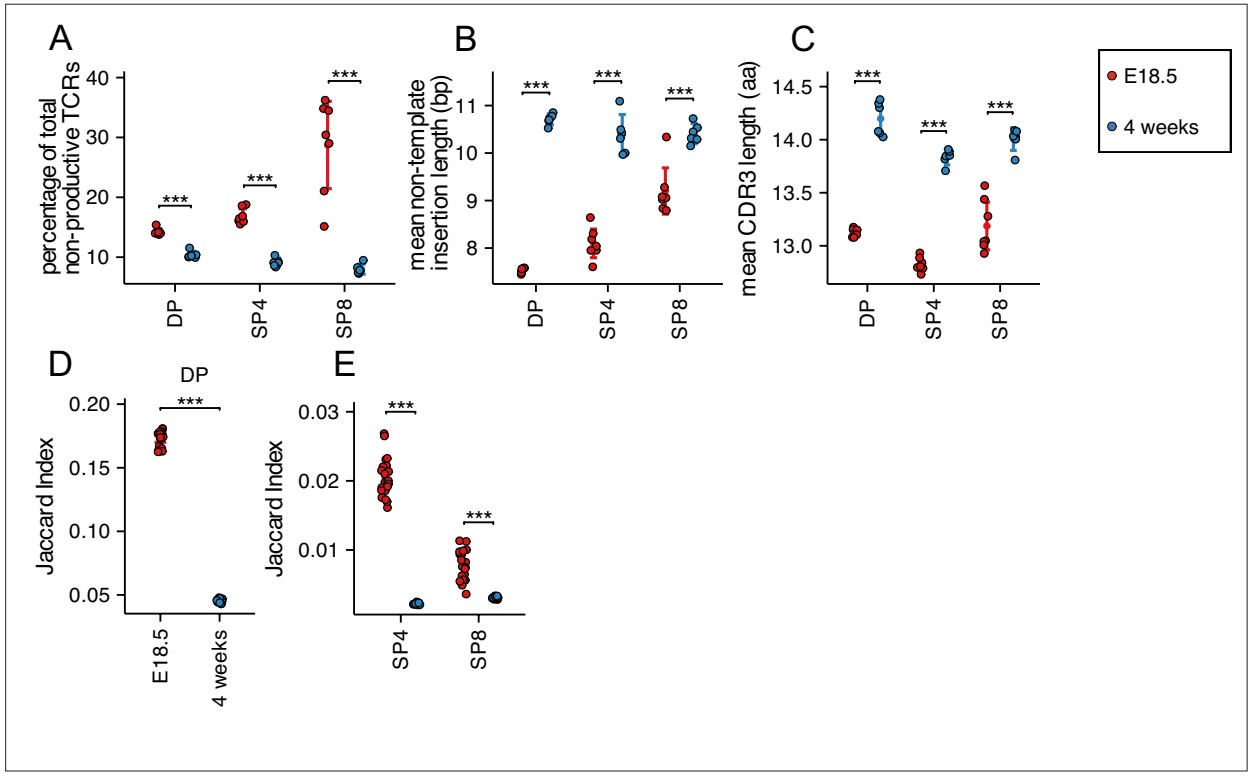

**Figure 5.** Decreased TCRβ non-template insertions, CDR3 length with increased CDR3 sharing, but reduced amino acid motif sharing in foetal compared to young adult. TCR β-chain repertoires were sequenced from FACS-sorted CD4⁺CD8⁺ (DP), CD4⁺CD8⁻CD3⁺ (SP4) and CD4⁻CD8⁺CD3⁺ (SP8) thymocyte populations from E18.5 (red circles; n=7) and 4 week-old (blue circles; n=6) C57BL/6 thymus. (**A**) Percentage of non-productive β-chain TCRs from total rearrangements for E18.5 and 4 weeks in DP, SP4 and SP8 populations. (**B**) The weighted mean of non-template insertion length for all β-chain TCRs (basepairs) for E18.5 and 4 weeks in DP, SP4 and SP8 populations. (**C**) The mean of β-chain unique predicted CDR3 length (number of amino acids) for E18.5 and 4 weeks in DP, SP4 and SP8 populations. (**D**) The intra life-stage Jaccard Index of Similarity of β-chain CDR3s for E18.5 and 4 weeks in the DP population. Each repertoire was subsampled to 30,000 CDR3s 1000 times before calculating the Jaccard Index. (**E**) The intra life-stage Jaccard Index of Similarity of β-chain CDR3s for E18.5 and 4 weeks in the SP4 and SP8 population. Each repertoire was subsampled to 500 CDR3s 1000 times before calculating the Jaccard Index. Dotplots show mean ±c.i and statistical comparisons were carried out by unpaired Student's t-test or Welch's t-test as appropriate: *** p<0.001; ** p<0.01; * p<0.05.

The online version of this article includes the following figure supplement(s) for figure 5:

**Figure supplement 1.** TCRα CDR3 non-template insertions, length, sharing and PCA of α-chain CDR1xCDR2 frequency distributions in foetal and young adult thymocyte populations.

to adult, indicating that the TCRβ CDR3 repertoires in each population from different embryos shared more sequences than the CDR3 repertoires from different individual adults (*Figure 5D and E*). This increased sharing of TCRβ CDR3 sequences in the foetal repertoires is consistent with shorter CDR3 length and shorter non-template insertions, leading to a repertoire that contains more genomic sequence than adult.

In the case of the α-chain repertoires, foetal DP and SP4 repertoires shared more TCRα CDR3 sequences between individuals than adults shared between individuals (*Figure 5—figure supplement 1D and E*).

## Summary

Foetal TCRβ and TCRα CDR3 repertoires have lower mean non-template insertion length and lower mean CDR3 length than adult, and share more CDR3 sequences with one another than adult.

## Different CDR1 and CDR2 usage between foetal and adult TCRβ and TCRα repertoires: less MHC-restriction in foetal repertoires

The CDR1 and CDR2 loops in the TCR are encoded in the V gene segments and typically contact the MHC's conserved α-helices. They can determine MHC restriction during repertoire selection (*Wong et al., 2019*), so we examined CDR1xCDR2 combinations in the foetal and adult thymocyte populations. We carried out PCA, using the frequency distributions (counts) of each β-chain CDR1xCDR2 combination (corresponding to V gene segment) as input (*Figure 6A–C*). The PCA clustered the populations by life-stage and cell-type, with all adult samples falling on the negative side of PC1, and clustering tightly by cell type on PC2, so that adult SP8 repertoires were positive on PC2, and adult DP and SP4 populations negative on PC2, with adult DP positioned between adult SP clusters. In contrast the foetal SP8 population was more dispersed and failed to cluster tightly on PC2, and DP repertoires fell on the positive side of PC2. To further visualise this, we generated a heatmap of the proportional usage of β-chain CDR1xCDR2 combinations for each foetal or adult thymocyte population (*Figure 6D*). The foetal populations clustered together, showing preference for distinct CDR1xCDR2 combinations, which reflect the bias observed in their TRBV gene usage (*Figure 2C*). For both foetal and adult samples, DP populations were positioned between SP8 and SP4, indicating a divergence in CDR1xCDR2 usage as a result of MHC restriction following positive selection. Indeed, CDR1xCDR2 combinations that contributed strongly to the positive side on PC2 (*Figure 6C*), such as MSHET-SYDVDS (TRBV29) and SGHSN-HYEKVE (TRBV12-1), showed increased expression in SP8 and decreased expression in SP4 (consistent with MHCI-restriction), while combinations that contributed strongly to the negative side of PC2, such as GKSSPN-SITVG (TRBV31), showed increased expression in SP4 and decreased expression in SP8 (consistent with MHCII-restriction).

TCRα CDR1 and CDR2 are also encoded in V gene segments and may also be determined by MHC-restriction (*Sim et al., 1996*), so we additionally investigated TCRα CDR1xCDR2 combinations (*Figure 6E*). Again, samples from each life-stage grouped together, and consistent with the strong life-stage dependent influence on TRAV use, several CDR1xCDR2 combinations showed an adult-bias in usage. PCA, using the frequency distributions (counts) of each predicted α-chain CDR1xCDR2 combination as input clustered adult repertoires on the negative axis of PC1, and separated adult SP4 repertoires from adult SP8 on PC2 indicating that TCRα CDR1/CDR2 usage was influenced by MHC restriction in adult repertoire selection, consistent with other reports in mouse and human (*Camaglia et al., 2023*; *Suo et al., 2024*). In contrast, foetal repertoires were highly dispersed and failed to segregate on either PC1 or PC2 (*Figure 5—figure supplement 1F and G*).

## Summary

CDR1xCDR2 (V gene segment) usage is different between adult and foetal repertoires, with adult TCRβ and TCRα CDR1xCDR2 repertoires clustering more tightly by PCA and segregating according to cell type (MHC-restriction).

## Distinct MHC-restriction between adult and foetal TCR repertoires

PCA of β-chain VxJ, CDR1βxCDR2β (Vβ), and CDR1αxCDR2α (Vα), separated adult but not foetal SP repertoires according to cell type, suggesting a weaker influence of MHC-restriction on the foetal single positive repertoires. However, foetal and adult TCR repertoires show bias to usage of different

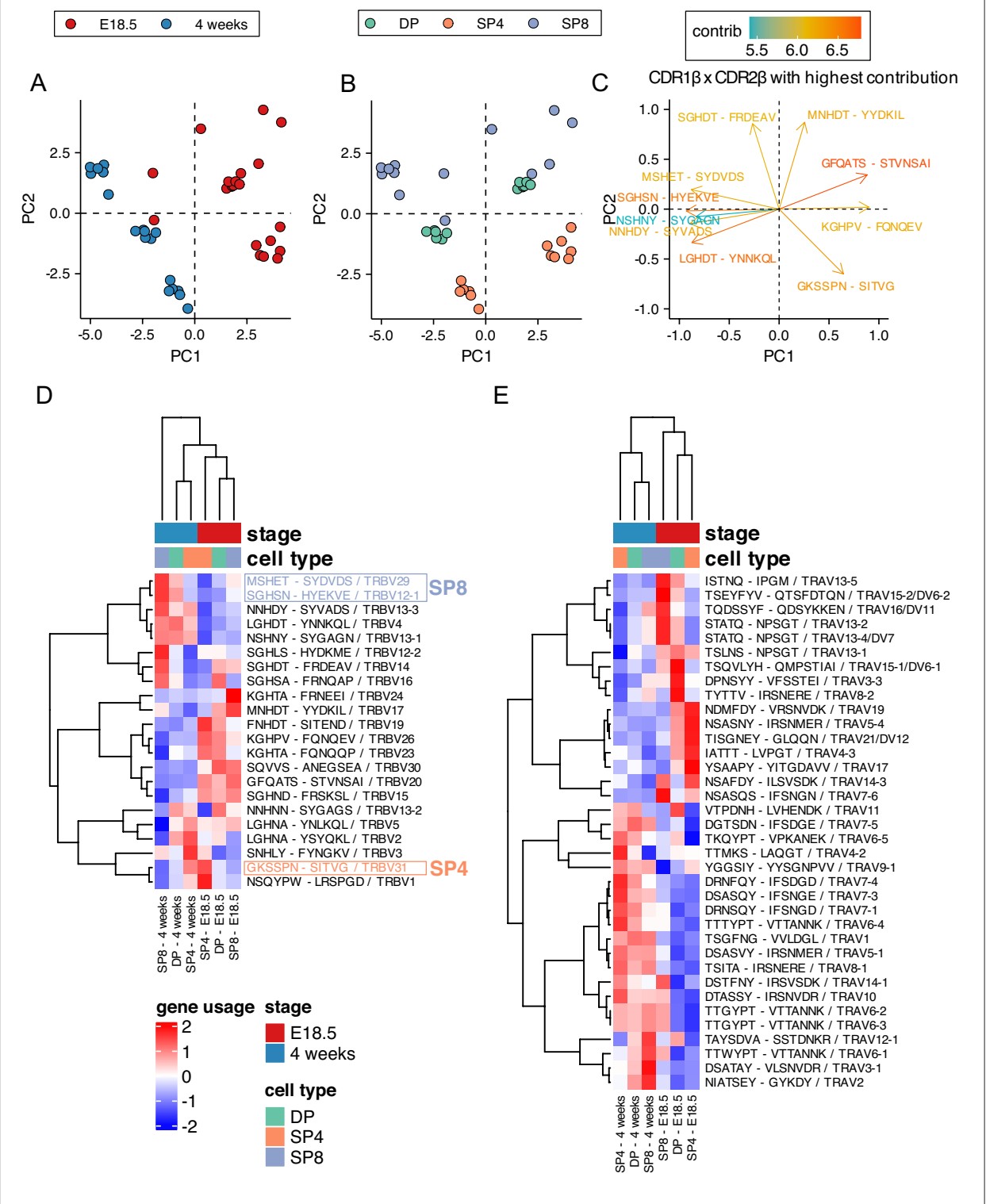

**Figure 6.** Proportional combinatorial CDR1xCDR2 bias in foetal and young adult thymocyte populations indicates reduced MHC-restriction in foetal thymus. TCR β-chain and α-chain repertoires were sequenced from FACS-sorted CD4⁺CD8⁺ (DP), CD4⁺CD8⁻CD3⁺ (SP4), and CD4⁻CD8⁺CD3⁺ (SP8) thymocyte populations from E18.5 (n=7) and 4-week-old (n=6) C57BL/6 thymus and frequency of CDR1 and CDR2 sequences calculated for each population. E18.5 and 4-week samples are coloured red and blue respectively, while DP, SP4, and SP8 populations are coloured green, orange, and purple respectively. (**A, B**) PCA biplot of β-chain CDR1xCDR2 frequency distributions for E18.5 and 4 weeks in DP, SP4, and SP8 populations. In (**A**), E18.5 and 4 weeks samples are coloured in red and blue respectively, while in (**B**), DP, SP4, and SP8 populations are coloured in green, orange, and purple respectively. (**C**) Top 10 highest CDR1xCDR2 combinations that contribute to PC1 and PC2 of the PCA of β-chain CDR1xCDR2 frequency distributions for

*Figure 6 continued on next page*

*Figure 6 continued*

E18.5 and 4 weeks in DP, SP4, and SP8 population (shown in A-B). (**D**) Heatmap of proportional β-chain CDR1xCDR2 usage E18.5 and 4 weeks in DP, SP4, and SP8 populations. Each column represents a mean of seven embryos or six adult mice and was clustered using Euclidian distance, while rows (CDR1 x CDR2 combinations) were clustered using Pearson correlation. (**E**) Heatmap of proportional α-chain CDR1xCDR2 usage in E18.5 and 4-week samples in DP, SP4, and SP8 populations. Each column represents a mean of seven embryos or six mice and was clustered using Euclidian distance, while rows (CDR1xCDR2 combinations) were clustered using Pearson correlation.

V and J gene segments (*Figure 2A–D*, *Figure 2—figure supplements 1 and 2*), so it is possible that foetal repertoires would cluster by cell type if PCA was carried out using foetal repertoires alone. We therefore repeated the PCA analyses on foetal and adult samples separately (*Figure 7A–L*). For foetal β-chain VxJ combinations, DP repertoires were separated from both SP repertoires on PC1 (corresponding to maturation), whereas SP4 repertoires separated from SP8 repertoires on PC2 (corresponding to MHC-restriction), which accounted for 11.1% of the variance between repertoires (*Figure 7A*). In contrast, for adult repertoires SP4 repertoires were separated from SP8 repertoires on PC1, which accounted for 44.2% of variance (*Figure 7B*). Thus, MHC-restriction accounted for approximately 4-fold more of the variance in β-chain VxJ counts in the adult repertoires than in foetal repertoires. For the adult repertoires, the 10 β-chain VxJ combinations that contributed most to PC1 (MHC-restriction) included combinations that contributed to both the positive (SP4) and negative (SP8) side of PC1; whereas for foetal repertoires the top 10 β-chain VxJ combinations all contributed to the positive side of PC1 (DP cells; *Figure 7C and D*).

PCA of foetal α-chain VxJ counts failed to cluster repertoires by cell type on either PC1 or PC2 (*Figure 7E*). In contrast, PCA of adult α-chain VxJ counts separated repertoires by SP cell type on PC1, which accounted for 25.9% of variance, with all SP4 repertoires falling on the positive side of the axis, and SP8 repertoires on the negative side (*Figure 7F*).

We also carried out PCA of β-chain and α-chain CDR1xCDR2 (V region) frequency distributions for adult and foetal repertoires separately. For both adult and foetal repertoires, PCA separated β-chain repertoires by SP cell type (MHC-restriction) on PC1 (*Figure 7G and H*). PC1 accounted for 56.1% of variability in the adult data, but only 28.8% of variance in foetal data, indicating a stronger impact of MHC-restriction on β-chain V region usage in adult than foetal thymocytes. Some combinations that contributed strongly to the negative (SP4) side of PC1 in the adult PCA, such as GKSSFN-SITVG (TRBV31), also contributed strongly to the foetal SP4 axis, but top contributors to SP4 in each PCA also included life-stage specific CDR1xCDR2 combinations, such as LGHNA-YSYQKL (TRBV2) and LGHNA-YNLKQL (TRBV5) for adult repertoires, and NSQYPQ-LRSPGD (TRBV1) for foetal repertoires (*Figure 7I and J*). Several CDR1xCDR2 contributed strongly to the SP8 axis in both adult and foetal PCA, such as MNHDT-YYDKIL (TRBV17) and MSHET-SYDVDS (TRBV29), but life-stage specific combinations (KGHTA-FRNEEI [TRBV24] for foetal SP8 and SGHLS-HYDKME [TRBV12-2] for adult SP8) also contributed strongly. Thus, for both SP4 and SP8 populations, life-stage influences which V region genes are selected.

PCA of α-chain CDR1xCDR2 (V region) counts for adult repertoires again separated SP4 repertoires from SP8 repertoires on PC1, accounting for 39.2% of variance, but failed to separate foetal repertoires by SP cell type on either PC1 or PC2 (*Figure 7K and L*), confirming a stronger impact of MHC-restriction on the adult than foetal TCRα repertoire.

### Summary

MHC-restriction contributes more to the variance in adult than foetal TCRβand TCRα repertoires. This indicates that V gene region usage and VxJ combinatorial usage are more tightly governed by MHC-restriction in adult than in foetal thymus.

## Young adult DP VJα gene segment usage becomes more foetal-like after hydrocortisone treatment

We observed distinct patterns of V and J gene usage between foetus and young adult, with foetal DP repertoires showing bias towards use of 3' V and 5' J rearrangements (*Figures 2–4*). During TCRβ gene rearrangement, these segments are closest to one another in the looping structure that enables V to DJ rearrangements (*Skok et al., 2007*), suggesting this might account for the foetal bias in proportional TRBVxTRBJ segment usage. In contrast to *Tcrb* locus rearrangement, the *Tcra* locus can

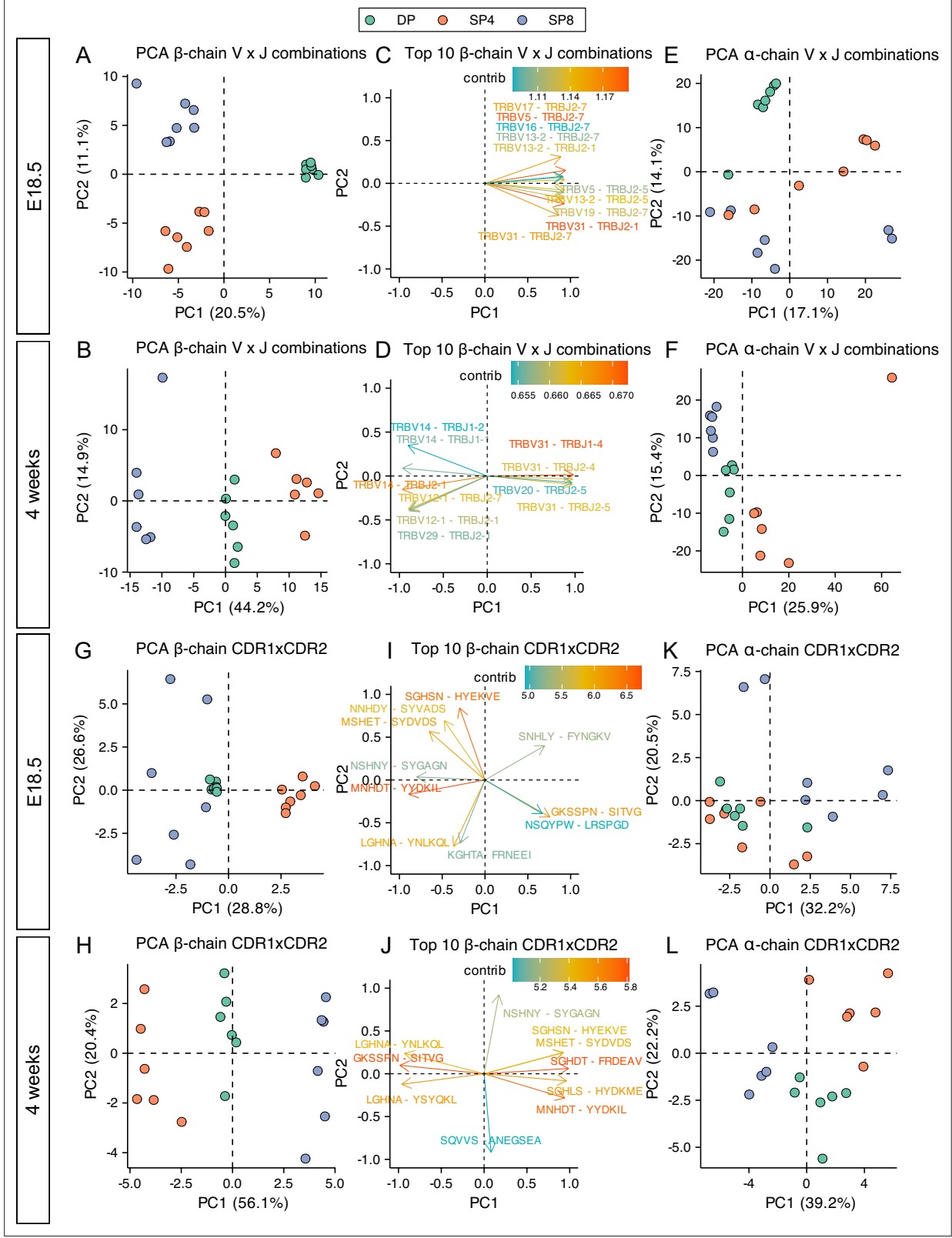

**Figure 7.** PCA of VxJ and CDR1xCDR2 combinations in foetal and young adult thymocyte populations indicate reduced MHC-restriction of foetal TCR repertoires. TCR β-chain and α-chain repertoires were sequenced from FACS-sorted CD4⁺CD8⁺ (DP), CD4⁺CD8⁻CD3⁺ (SP4) and CD4⁻CD8⁺CD3⁺ (SP8) thymocyte populations from E18.5 (n=7) and 4-week-old (n=6) C57BL/6 thymus. (**A, B**) PCA biplot of β-chain VxJ gene counts of unique TCRβ sequences for E18.5 (**A**) and 4 week (**B**) in DP, SP4, and SP8 populations (coloured in green, orange, and purple respectively). The percentage of

*Figure 7 continued on next page*

*Figure 7 continued*

variance for each principal component is indicated on the axis. (**C**) Ten β-chain VxJ gene combinations that contribute most to PC1 and PC2 of the PCA for E18.5. (**D**) Ten β-chain VxJ gene combinations that contribute most to PC1 and PC2 of the PCA for 4 weeks. (**E, F**) PCA biplot of α-chain VxJ gene counts of unique TCRβ sequences for E18.5 (**E**) and 4 weeks (**F**) in DP, SP4, and SP8 populations (coloured in green, orange, and purple respectively). The percentage of variance for each principal component is indicated on the axis. (**G, H**) PCA biplot of β-chain CDR1xCDR2 frequency distributions of unique TCRβ sequences for E18.5 (**G**) and 4 weeks (**H**) in DP, SP4, and SP8 populations (coloured in green, orange, and purple respectively). The percentage of variance for each principal component is indicated on the axis. (**I**) Ten β-chain CDR1xCDR2 combinations that contribute most to PC1 and PC2 of the PCA of β-chain CDR1xCDR2 frequency distributions for E18.5. (**J**) Ten β-chain CDR1xCDR2 combinations that contribute most to PC1 and PC2 of the PCA of β-chain CDR1xCDR2 frequency distributions for 4 weeks. (**K, L**) PCA biplot of α-chain CDR1xCDR2 frequency distributions of unique TCRβ sequences for E18.5 (**K**) and 4 weeks (**L**) in DP, SP4, and SP8 populations (coloured in green, orange, and purple respectively). The percentage of variance for each principal component is indicated on the axis.

undergo multiple rounds of rearrangement on each chromosome, sequentially moving 3' to 5' along the series of TRAV gene segments, while moving 5' to 3' along the series of TRAJ gene segments, with proximal pairs (3' TRAV segments with 5' TRAJ segments) initiating this sequence of rearrangements (*Carico et al., 2017*). We observed that the foetal DP repertoire showed TRAVxTRAJ bias by their chromosomal position. One possible explanation for this bias is that in the foetus progressive rounds of *Tcra* rearrangement are less common than in young adult, perhaps as a result of differences in the kinetics of T-cell development. Differentiation of foetal thymocytes occurs in a largely synchronized wave, whereas adult thymocyte production is unsynchronised, continuous and slower. Once the adult thymus has reached steady-state, homeostasis between thymocyte populations regulates the rate of differentiation to DP cell (*Hager-Theodorides et al., 2007*; *Outram et al., 2009*). To examine the influence of the rate of differentiation on VJ gene usage, we synchronized the differentiation of adult DP thymocytes by treating young adult mice with hydrocortisone (HC) to deplete the adult thymus of all but the most mature cells. At 2 days after treatment, the thymus was depleted of DP cells. Four days later (6 days after treatment), we FACS-sorted and TCR sequenced the replenished CD3$^{-/lo}$DP (CD3$^{-/lo}$CD4$^+$CD8$^+$), and CD3$^{+/hi}$DP (CD3$^{+/hi}$CD4$^+$CD8$^+$) populations. We plotted heatmaps of mean proportional V and J gene segment usage for unique TCR sequences in foetal, HC-treated and control young adult repertoires, clustering each life-stage and thymocyte population, but showing each gene segment in chromosomal order (*Figure 8A–D*). For TRAJ usage, foetal and adult HC-treated repertoires clustered together, with untreated adult repertoires clustering separately (*Figure 8A and B*). Interestingly, both foetal and HC-treated young adult populations favoured 5' TRAJ segments, in contrast to control young adult populations, that displayed bias for 3' TRAJ (*Figure 8A and B*). We next compared usage of all possible VxJ combinations between control and HC-treated young adult DP populations for all *Tcra* sequences, identifying combinations that showed a change in proportional usage in the HC-treated young adult compared to control young adult (*Figure 4A–C*). The DP HC-treated young adult population favoured VxJ combinations from the 3' location of TRAV and 5' location of TRAJ along with clear proportional decrease in usage of combinations of 5' TRAV and 3' TRAJ in comparison to young adult control (*Figure 8C*). Therefore, by synchronizing young adult thymus using HC, foetal, and young adult gene segment usage became more alike, with 5' TRAV and 3' TRAJ bias (*Figures 4A and 8C–F*).

In the case of the β-chain, comparison of proportional usage of all possible unique TCRβ VxJ combinations between HC-treated and control young adult DP populations showed a 5' bias in TRBJ gene segment usage (*Figure 8F*), with several VxJ combinations that were proportionally increased or decreased in common with the comparison between foetal and young adult combinatorial gene usage (*Figures 4A and 8F*). These data indicate that when the adult young adult DP population arises quickly and synchronously its TRAV, TRAJ and TRBJ segment usage and TRAVxTRAJ combinations closely resemble the foetal DP repertoire, suggesting that the kinetics of differentiation are important in determining gene segment usage.

## Summary
As the young adult thymus recovers from hydrocortisone depletion, combinatorial TCRα VxJ usage becomes more foetal-like, suggesting that the synchronicity and kinetics of foetal T-cell development influence VxJ combination.

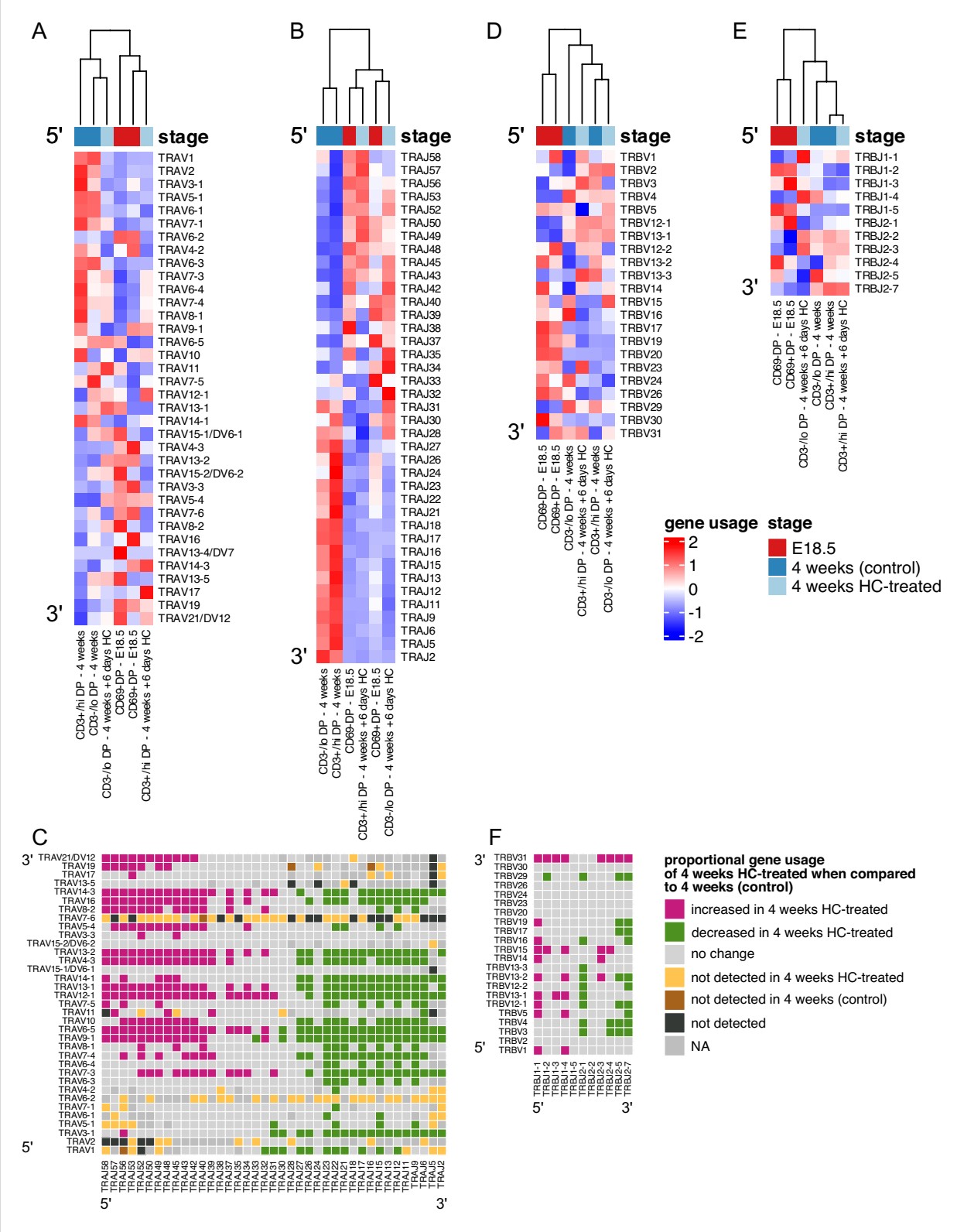

**Figure 8.** In hydrocortisone-depleted adult thymus recovering DP populations use foetal-like gene segments. TCR α-chain (**A–C**) and β-chain (**D–F**) repertoires were sequenced from FACS-sorted CD4+CD8+CD69- (CD69-DP, foetal), CD4+CD8+CD69+ (CD69+DP foetal) CD4+CD8+CD3-/lo (CD3-/loDP adult), CD4+CD8+CD3+/hi (CD3+/hiDP adult) thymocyte populations from E18.5 (red; n=7), 4-week-old (blue; n=6), and 4-week-old treated with hydrocortisone on day 6 after treatment (light blue; n=5) C57BL/6 thymus. (**A, B**) Heatmaps of proportional α-chain variable (V) (**A**) and α-chain joining

*Figure 8 continued on next page*

*Figure 8 continued*

(J) (**B**) gene usage of productive TCRs for E18.5, 4 weeks (control) and 4 weeks treated with hydrocortisone in DP populations. Each column represents a mean of 7 or 6 (E18.5 and 4 weeks) or 5 (4 weeks treated with hydrocortisone) embryos or mice and was clustered using Euclidian distance. Genes are shown in chromosomal order (5′ to 3′) from top to bottom. (**C**) Plot shows α-chain proportional VxJ gene usage of total TCRs for 4 weeks HC-treated compared to 4 weeks (control) in DP populations. Pink tiles signify increased usage of VxJ combinations in 4 weeks HC-treated (p<0.05), while green tiles signify decreased usage in 4 weeks HC-treated (p<0.05) compared to 4 weeks (control). Grey tiles signify no change in gene usage (p>0.05), yellow tiles signify VxJ combinations not detected in 4 weeks HC-treated, brown tiles signify VxJ combinations not detected in 4 weeks (control) and black tiles signify VxJ combinations not detected in both 4 weeks HC-treated and 4 weeks (control). Only combinations that were detected in at least three mice per group were compared. Dark grey tiles signify combinations that were not compared. Statistical comparisons were carried out by unpaired Student's t-test followed by FDR-adjustment (5%, Benjamini-Hochberg procedure) of p values. (**D, E**) Heatmaps of proportional β-chain V (**D**) and β-chain J (**E**) gene usage of unique TCRs for E18.5, 4 weeks (control) and 4 weeks treated with hydrocortisone in DP populations. Each column represents a mean of 7 or 6 (E18.5 and 4 weeks) or 5 (4 weeks treated with hydrocortisone) embryos or mice and was clustered using Euclidian distance. Genes are shown in chromosomal order (5′ to 3′) from top to bottom. (**F**) Plot shows β-chain proportional VxJ gene usage of total TCRs for 4 weeks HC-treated compared to 4 weeks (control) in DP populations. Pink tiles signify increased usage of VxJ combinations in 4 weeks HC-treated (p<0.05), while green tiles signify decreased usage in 4 weeks HC-treated (p<0.05) compared to 4 weeks (control). Grey tiles signify no significant change in gene usage (p>0.05), yellow tiles signify VxJ combinations not detected in 4 weeks HC-treated, brown tiles signify VxJ combinations not detected in 4 weeks (control) and black tiles signify VxJ combinations not detected in both 4 weeks HC-treated and 4 weeks (control). Statistical comparisons were carried out by unpaired Student's t-test followed by FDR-adjustment (5%, Benjamini-Hochberg procedure) of *p* values.

## Discussion

Here, we employed a bulk population-based strategy to investigate the TCRβ and TCRα chain repertoires from developmentally defined thymocyte populations during T-cell development. Our study revealed many differences between the foetal and young adult thymus in TCR gene segment usage, repertoire diversity, and clonality by bulk sequencing separate TCRβ and TCRα repertoires. These data indicate that MHC restriction and positive selection have a weaker impact on the TCR repertoire in foetal thymus compared to adult. Overall, the differences between the foetal and adult thymus TCR repertoires are consistent with the foetal thymus producing αβT-cells with properties and functions that are distinct from adult T-cells: their repertoire is less diverse, more closely encoded by genomic sequence, less governed by MHC-restriction, and with preference for particular gene segment usage. Several recent studies have demonstrated distinct and more innate-like features of foetal and neonatal T-cell populations in both humans and mice (*Beaudin et al., 2016*; *Wang et al., 2016*; *Rackaityte and Halkias, 2020*; *Smith et al., 2018*). The first wave of foetal αβT-cells that leave the thymus must provide early protection against infection in the neonatal animal but also need to be tolerant to both self and maternal MHC/antigens. It is possible that the distinct features and potentially weaker MHC-restriction of the foetal TCR repertoire have evolved to provide these two important features of foetal adaptive immunity. It will be interesting to investigate if the foetal-bias in gene segment usage confers specific properties on foetal TCRs, with respect to their binding specificity, affinity and avidity and immunity to neonatal infection (*Thomas and Crawford, 2019*).

In validation of our approach, we were able to quantify known differences in TCRβ repertoire diversity, CDR3 length and non-template insertion length between foetal and adult (*Sethna et al., 2017*). The foetal TCRβ and TCRα repertoires contained fewer non-template nucleotides than adult, indicating that they were more closely aligned to genomic sequence. Consistent with this, the foetal β-chain and α-chain CDR3 repertoires shared more sequences with one another than their young adult counterparts. Our study also showed that the foetal TCRβ repertoire had a less equal distribution in DP, SP4, and SP8 populations compared to adult, and increased clonal expansion of the most abundant clones in DP and SP4 populations. Thus, the reduction in TCRβ repertoire diversity observed in the foetal thymus could not simply be attributed to the reduction in N-nucleotide insertions. These differences in TCRβ repertoire distribution and clonal expansion may be the result of differences in the kinetics of expansion and differentiation of thymocyte subsets in the foetus, leading to greater clonal expansion of TCRβ clones following β-selection. The transition from DP to SP cell in the foetus takes just 2–3 days at most, as DP cells first appear around E16, and the mature SP populations were FACS-sorted on E18.5, whereas in the adult thymus differentiation from DN3 to SP cell may take place over a longer period of time (*Ross et al., 2014*; *Solanki et al., 2020*).

We found many differences in proportional TRBV, TRBJ, TRAV, and TRAJ gene usage between foetal and adult thymocyte populations, with foetal populations showing proportional bias towards 3′ TRBV usage. We observed a particularly striking bias towards use of 3′ TRAV and 5′ TRAJ gene

segments in the foetus. As a previous study used a PCR-based approach to identify enrichment of 3'TRAV to 5'TRAJ rearrangements in foetal BALB/c thymocytes (*Pasqual et al., 2002*), the bias is not dependent on mouse strain or MHC haplotype.

Bias in TCRα gene segment usage by chromosomal location has been reported in human DN thymocyte populations by single cell RNAseq but was found to diminish as thymocytes mature and progressive recombination of the TCRα loci occurs (*Park et al., 2020*). In contrast, in the mouse foetal thymus, we found that the bias in proportional TRAV and TRAJ gene segment use by chromosomal position persisted in the positively selecting CD69[+]DP and mature SP4 populations, but was not apparent in the adult CD3[-/lo]DP population. Our data therefore suggest that progressive rounds of TCRα rearrangement are less common in the foetal DP population. Indeed, when we depleted the young adult thymus by HC-treatment, we found an increase in 3'TRAV and 5'TRAJ gene segment usage in the synchronized recently differentiated recovering DP population, making these repertoires more similar to the foetal DP TCRα repertoires. Thus, differences between foetal and control young adult TCRαV-J rearrangements in DP cells may be the result of slower differentiation in the adult thymus, allowing more time for multiple rounds of TCRαV-J rearrangements in the DP population. Consistent with this, the foetal DP population showed a different TCRα frequency distribution than the adult DP population, with a lower power law exponent.

Comparison of proportional use of TCRβVxJ and TCRαVxJ combinations between adult and foetal populations also highlighted the 3' V gene segment preference in the foetal DP population, and the 5' V gene segment preference in the adult DP population for both β-chain and α-chain. Interestingly, life-stage had a greater impact on TCRβ and TCRα gene segment usage than MHC-restriction in all populations.

When we carried out PCA on adult and foetal samples separately, PCA clustered β-chain VxJ combination counts and β-chain CDR1xCDR2 by SP population in both adult and foetal samples, showing the impact of MHC restriction (positive selection) on the SP repertoires. However, the percentage of variability that was attributable to MHC-restriction was greater in adult repertoires than foetal repertoires. PCA also clustered the adult α-chain V x J and CDR1xCDR2 combinations by SP cell type, confirming impact of MHC restriction on adult Vα gene usage. However, foetal populations failed to segregate by cell type in either α-chain PCA, indicating that MHC restriction/positive selection has less influence on α-chain repertoires in foetal SP populations. Thus, our data suggests that in the foetal thymus the influence of MHC-restriction/positive selection on both the TCRβand TCRα repertoires that is weaker than in the adult thymus.

Our approach of bulk TCR sequencing from FACS-sorted thymocyte populations had the advantages of scale (allowing the sequencing of thousands of rearrangements from thousands of cells from each embryo or mouse), at relatively low cost. However, it had the disadvantage that we were unable to investigate which TCRβ rearrangements pair with which TCRα rearrangements in single cells. Clearly, in future it will be important and interesting to expand this analysis to include TCRβ/α pairing by single cell RNAseq.

## Materials and methods
### Mice
C57BL/6 mice were bred and maintained in specific pathogen-free conditions at University College London (UK) and animal experimentation was carried out under UK Home Office regulations (PPL PP7936049), following ethical review at UCL, following ARRIVE guidelines. Mixed sex of foetal (E18.5) and young adult (4 weeks old) mice were used. For hydrocortisone (HC) treatment, mice were injected intraperitoneally with 0.6 mg/g of body weight pure HC sodium phosphate (Sigma-Aldrich, cat. no: BP188) in sterile PBS, and analysed after 6 days.

### Fluorescence activated cell sorting
Each foetal or young adult thymus was disaggregated into a single cell solution using the back of a syringe on a 70-µn cell strainer. The cells were washed through the filter into falcon tubes using ice cold FACs buffer (1 x AIM-V medium (research grade), AlbuMAX Supplement (Gibco, cat. no: 31035025)) to eliminate any large clumps. Cells were then counted using Accuri C6 flow cytometer and then pelleted by centrifugation for 5 minutes at 1400 rpm and supernatant was removed. The

whole cell pellet from the entire organ was stained in FACS buffer with a panel of directly conjugated antibodies supplied by Biolegend (San Diego, US) or eBioscience (San Diego, US; see *Supplementary files 1 and 2*) on ice for 30 min as described (*Lau et al., 2021*). To obtain CD3$^{-/lo}$DP (CD4$^+$CD8$^+$CD3$^{-/lo}$), CD69$^-$DP (CD4$^+$CD8$^+$CD69$^-$), CD69$^+$DP (CD4$^+$CD8$^+$CD69$^+$), CD3$^{+/hi}$DP (CD4$^+$CD8$^+$CD3$^{+/hi}$), SP4 (CD4$^+$CD8$^-$CD3$^+$), SP8 (CD4$^-$CD8$^+$CD3$^+$) thymocyte cell suspensions were sorted using a BD FACS Aria III. CD69 expression was used to differentiate DP populations in the foetus, as anti-CD3 staining is dull in DP thymocyte populations at E18.5 (*Solanki et al., 2018*). Only live cells were gated and collected using forward and side scatter. Gating strategies to sort the required cell populations for experiments shown in *Figure 1* and all subsequent figures are shown in *Figure 9* (for foetal thymus) and *Figure 9—figure supplement 1* (for adult thymus).

The cells were collected in 100 µL (foetus) or 400 µL (young adult) of FACS buffer and were then pelleted by centrifugation for 25 min at 1400 rpm at 4 °C. The supernatant was removed and 100 µL of Extraction buffer from PicoPure RNA isolation kit (Applied Biosystems, cat. no. KIT0204) was added. Finally, the samples were incubated for 30 min at 42 °C on a shaking heatblock and then stored at –80 °C until the RNA was ready to be extracted.

## RNA extraction

RNA was extracted by PicoPure RNA isolation kit (Applied Biosystems, cat. no. KIT0204) according to manufacturer's instructions and eluted in 11 µL of elution buffer. RNA concentration was assessed using a spectrophotometer.

## TCR amplification and sequencing

We used the protocol for TCR amplification and sequencing as described (*Oakes et al., 2017*; *Uddin et al., 2019*), with some minor modifications for mouse samples. See *Supplementary files 3 and 4* for the primer list with the sequences and for a full list of reagents respectively.

RNA was first DNase treated. A maximum of 500 ng of RNA was mixed with 1 µL RQ1 10 x Buffer (Promega, cat. no: M6101) and 1 µL RQ1 DNase (1 U/µL, Promega, cat. no: M6101) and then incubated at 37 °C for 30 min. 1 µL of RQ1 DNase Stop Solution (Promega, cat. no: M6101) was then added and the mix was incubated at 65 °C to stop the reaction. The RNA was then reverse transcribed by adding 1.5 µL of 10 µM TRAC2 primer, 1.5 µL of 10 µM TRBC3 primer, and 1.5 µL of 10 mM dNTPs (Promega, cat. no: U1515) and incubating at 65 °C for 5 min. The reaction was then rapid cooled down on ice and 6 µL of 5 x FS Buffer (Invitrogen, cat. no: 18080044), 1.5 µL of 0.1 M DTT (Invitrogen, cat. no: 18080044), 1.5 µL of RNasin Ribonuclease Inhibitor (40 U/µL, Promega, cat. no: N2111) and 1.5 µL of SuperScript III RT (200 U/µL, Invitrogen, cat. no: 18080044) was added and it was incubated at 55 °C for 30 min, followed by 70 °C for 15 min. These reactions were purified by Minelute PCR purification (QIAGEN, cat. no: 28004) following the manufacturer's instructions and eluted in 10.5 µL nuclease-free water.

The cDNA was then ligated to the M13 ligation primer which is composed of the Illumina SP2 primer, an 8 base spacer and a 12 base Unique Molecular Identifier (UMI). This UMI consists of two random hexamers separated by the 8 base spacer. On the 5' end the primer is phosphorylated, and on the 3' end it is blocked with a Spacer C3 moiety to prevent primer concatemerization. The ligation mix consisted of: 10 µL cDNA, 3 µL of bovine serum albumin (BSA)/hexamine cobalt chloride (HCC) mixture (1 mg/mL BSA, 10 mM HCC), 3 µL of T4 RNA ligase buffer (NEB, cat. no: M0204S), 1 µL of 10 mM ATP (NEB, cat. no: M0204S), 1 µL of 10 µM M13 Ligation primer, 2 µL of T4 RNA Ligase (10 000 U/mL, NEB, M0204S), and 10 µL of PEG 8000 50% (NEB, cat. no: M0204S). The ligation mix was then incubated for at least 18 hr up to a maximum of 23 hr at 16 °C and then heat inactivated at 65 °C for 10 min.

70 µL of nuclease-free water was added to the ligation products. This was then purified by adding 50 µL of AMPure XP Beads (Beckman Coulter, cat. no: A63881) to the 100 µL ligation product, with some modifications to manufacturer's instructions: 300 µL of 80% ethanol was used for the washing steps and the product was eluted in 31 µL nuclease-free water.

The purified ligation product was then amplified in a PCR reaction consisting of 31 µL purified product, 10 µL of 5 x HF Buffer (NEB, cat. no: M0530L), 2.5 µL of 10 µM m- alpha-RC1 primer, 2.5 µL of 10 µM m-beta-RC1 primer, 2.5 µL of 10 µM SP2-M13 primer, 1 µL of 10 mM dNTPs (Promega, cat. no: U1515), and 0.5 µL of Phusion Polymerase (NEB, cat. no:M0530L). The m-alpha-RC1 and m-beta-RC1

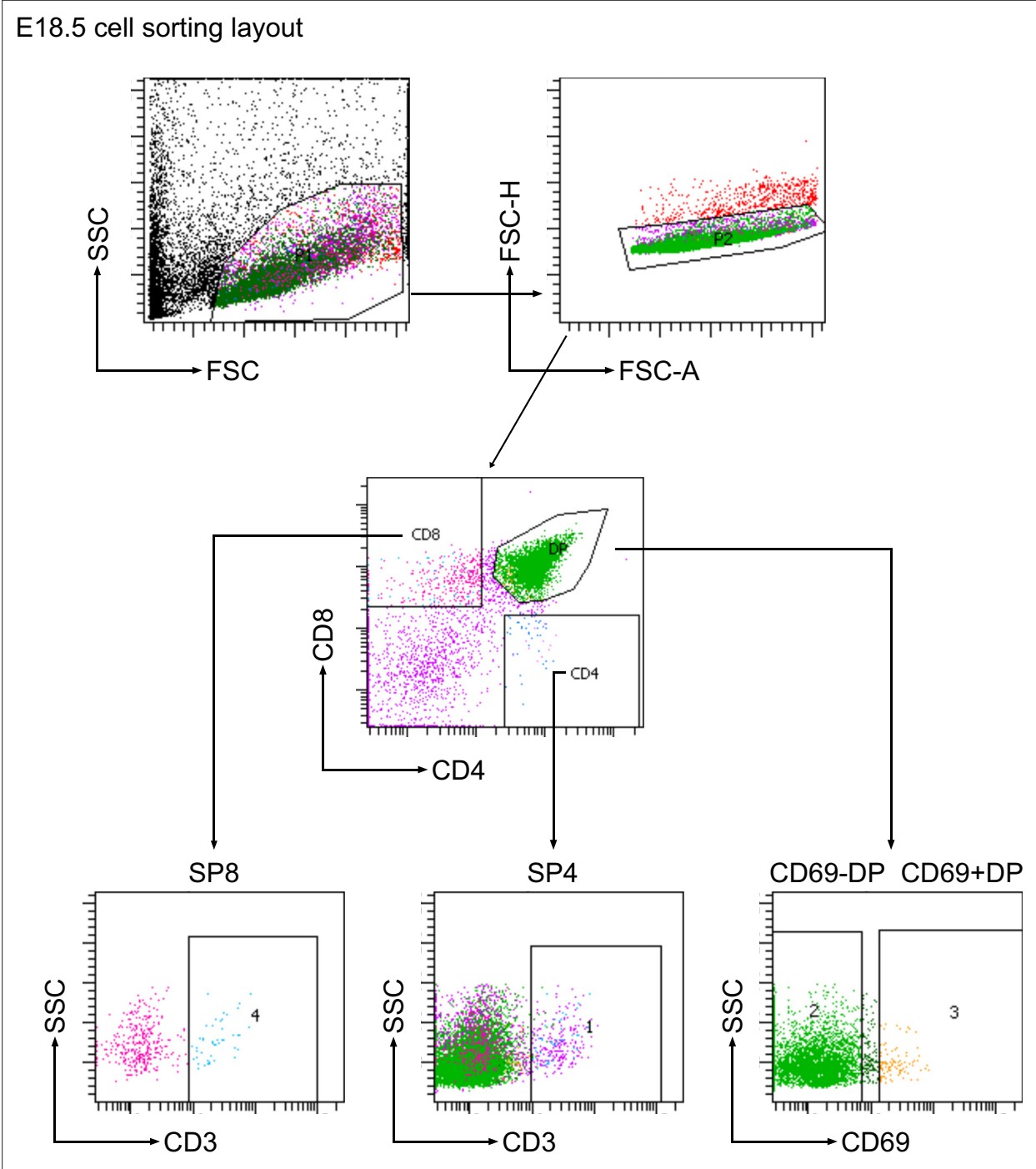

**Figure 9.** Cell sorting strategy for E18.5 thymus. Flow cytometry plots show the sequential gating strategy to sort E18.5 embryonic thymus populations by fluorescence activated cell sorting. Regions on the dot plots show the gates that were used.

The online version of this article includes the following figure supplement(s) for figure 9:

**Figure supplement 1.** Cell sorting strategy for adult thymus.

primers hybridize to the constant region of the α-chain and β-chain genes in the mouse, respectively. The SP2-M13 primer hybridizes to the sequence introduced by the M13 ligation primer earlier during the ligation step. PCR-1 was run using the following conditions: initial denaturation at 98 °C for 3 min, followed by 4 cycles of denaturation at 98 °C for 15 s, annealing at 69 °C for 30 ss and extension at 72 °C for 40 s, followed by a final extension at 72 °C for 5 min.

The 50 µL PCR product was then purified by adding 40 µL of AMPure XP Beads (Beckman Coulter, cat. no: A63881) with some modifications to manufacturer's instructions: 200 µL of 80% ethanol was used for the washing steps. At this stage, the samples were split into two: TCRα and TCRβ, each eluted in 31 µL nuclease-free water.

The purified PCR product was then amplified in a PCR reaction to add indices and Illumina adaptors P5 and P7 and sequencing primer 1 (SP1) sequence. The PCR reaction mix consisted of: 31 µL purified product, 2.5 µL of 1 µM SP1 index (mSP1- 6N-I-**X**-αRC1 or mSP1-6N-I-**X**-βRC1, for TCRα and TCRβ respectively), 2.5 µL of 10 µM SP2 index (P7-L**X**), 2.5 µL of 10 µM SP1-P5 primer, 10 µL of 5 x HF Buffer (NEB, cat. no: M0530L), 1 µL of 10 mM dNTPs (Promega, cat. no: U1515), and 0.5 µL of Phusion Polymerase (NEB, cat. no: M0530L). PCR-2 was run using the following conditions: initial denaturation at 98 °C for 3 min, followed by 6 cycles of denaturation at 98 °C for 15 s, annealing at 69 °C for 30 s and extension at 72 °C for 40 s, followed by a final extension at 72 °C for 5 min. The 50 µL PCR product was then purified by adding 40 µL of AMPure XP Beads (Beckman Coulter, cat. no: A63881) with some modifications to manufacturer's instructions: 200 µL of 80% ethanol was used for the washing steps, and the product was eluted in 30 µL nuclease-free water.

The purified PCR product was then amplified in a qPCR reaction, so that the reaction can be monitored and stopped in real time when the reaction passes the threshold of 0.1 ΔRn to prevent overamplification and minimise the introduction of PCR artifacts. SYBR Green I Nucleic Acid Gel Stain, 10,000 x concentrate in DMSO (ThermoFisher Scientific, cat. no: S7567) was diluted firstly 1:100 in DMSO and this stock solution was stored at –20 °C. Prior to the assembly of the PCR mix, this stock solution was diluted a further 1:50 in nuclease-free water.

The PCR reaction consisted of: 28 µL purified product, 10 µL of 5 x HF Buffer (NEB, cat. no: M0530L), 5 µL of SYBR Green (1:50), 1.25 µL of 10 mM dNTPs (Promega, cat. no: U1515), 1 µL of Rox (Invitrogen, cat. no: 12223012), 2.5 µL of 10 µM P5 primer, 2.5 µL of 10 µM P7 primer and 0.5 µL of Phusion Polymerase (NEB, cat. no: M0530L). The qPCR was run using the following conditions: initial denaturation at 98 °C for 3 min, followed by a variable number of cycles of denaturation at 98 °C for 15 s, annealing at 69 °C for 30 s and extension at 72 °C for 40 s, followed by a final extension at 72 °C for 5 min.

The reaction was stopped when the samples passed the threshold of 0.1 ΔRn. The 50 µL PCR product was then purified by adding 40 µL of AMPure XP Beads (Beckman Coulter, cat. no: A63881) with some modifications to manufacturer's instructions: 80% ethanol was used for the washing steps and the product was eluted in 30 µL nuclease-free water.

The purified products were then quantified using Qubit dsDNA high sensitivity reagents (ThermoFisher Scientific, cat. no: Q32854) and the size was confirmed with the TapeStation System HSD1000 (Agilent, cat. no: 5067–5584, cat. no: 5067–5585). The expected size is around 650 bp for a successful library preparation.

## Sequencing

The concentrations acquired by the Qubit and the size determined by the TapeStation, were used to calculate the molarity of the samples in nM. The samples were then diluted to 12 nM and pooled into one 12 nM library containing samples with different indices. 84 samples were prepared per sequencing run (42 α-chain and 42 β -chain libraries). A Vacuum Concentrator Centrifuge was used to pellet the library and remove excess volume, so that 30 µL of the pooled library could then be run on the Pippin Prep System using a Pippin Gel Cassette 1.5% w/v agarose dye free 250 bp-1.5 kb DNA size range (Sage Science, cat. no: CDF1510) to size select for sequences between 350 and 750 bp, following the manufacturer's instructions.

40 µL of library eluted from the Pippin Prep was then quantified using a Qubit and TapeStation System HSD1000 as detailed above. The expected molarity was between 2 nM and 5 nM. The library was then denatured and diluted to 1.2 pM according to standard Illumina's protocols (for 4 nM libraries) and was spiked with 22% of 20 pM PhiX control (Illumina, cat. no: FC-110–3001). The 1.2 pM library was sequenced on the NextSeq using the NextSeq 500/550 Mid Output Kit v2.5 (300 Cycles) (Illumina, cat. no: 20024905) at UCL genomics.

## Error correction and outputting

The NextSeq outputs files in the format named binary based call (.bcl) which were converted into FASTQ files using bcl2fastq for downstream processing using a pipeline of scripts described previously

(*Oakes et al., 2017*; *Thomas et al., 2013*): *Decombinator_v3.1* (available at: https://github.com/innate2adaptive/Decombinator/; copy archived at *innate2adaptive, 2024*) in Python 2.7. This pipeline identifies the errors introduced by PCR amplification using the UMI attached during the ligation step and thus produces an output of a corrected abundance of the TCR in the sample along with determining V, J and CDR3 regions for each TCR.

## Downstream analysis

For further downstream analyses, *R Studio* was used. The *tidyverse* set of packages were used for data manipulation and visualisations, in particular *ggplot2* (*Wickham, 2016*; *Wickham et al., 2019*). Dotplots show mean ±c.i (package *ggpubr* [*Kassambara, 2023a*]) and statistical comparisons were carried out by unpaired Student's t-test or Welch's t-test as appropriate using the package *rstatix* (*Kassambara, 2023bKassambara, 2023b*). *Pheatmap* and *ComplexHeatmap* were used to generate Pearson correlation heatmaps (*Gu et al., 2016*; *Kolde, 2019*).

## TCR abundance distribution

TCR repertoire abundance distributions are typically L-shaped, with a high number of distinct TCRs or clonotypes present only once, and a low number of hyperexpanded clones. Therefore, the TCR abundance (number of copies of each sequence) was plotted against their proportion of the repertoire in a log-log plot. This transformed distribution follows approximately a linear distribution apart from the largest clones and can be fitted to a discrete power law ($f(x) = kx^{-\alpha}$) as seen in previous studies (*Oakes et al., 2017*; *Joshi et al., 2019*). The TCR abundance frequency was fitted to a discrete power law using maximum likelihood estimation (*Bauke, 2007*; *Clauset et al., 2009*) using the *PoweRLaw* package (*Gillespie, 2015*). The power law exponents (α) were then plotted and compared.

## Diversity indices

Before calculating diversity indices, α and β identified TCRs or CDR3s were subsampled (rarefied) to a number lower than the smallest repertoire using the package *vegan* (*Oksanen et al., 2022*) in order to correct for differences in diversity due to sample size. The mean diversity indices (Shannon Entropy, Gini Index and Jaccard Index of Similarity) were then calculated from 1000 repeats of this random sampling. The Shannon Entropy was computed using the package *vegan* (*Oksanen et al., 2022*). Gini index was computed using *ineq* package (*Zeileis, 2014*). The base R function *dist* was used to calculate the Jaccard Index of Similarity.

## Principle component analysis

Before PCA was applied, raw counts were first $\log_{10}$ transformed to account for differences in sample size, using a pseudocount of 0.01 for any counts that were not observed. Z-scores were then calculated from these values by subtracting the mean and dividing by the standard deviation. PCA was then computed using the base R function *prcomp* and *factoextra* package was used to investigate the results (*Kassambara, 2020*).

## VxJ combinations

Proportional VxJ gene usage of total TCRs was calculated for each sample and only VxJ combinations that were detected in all samples were compared. Statistical comparisons were carried out by unpaired Student's t-test followed by FDR-adjustment (5%, Benjamini-Hochberg procedure) of *p* values.

## Acknowledgements

This work was funded by the MRC (MR/P000843/1, MR/S037764/1) and BBSRC (BB/T020970/1). JR was supported by a studentship from the BBSRC London Interdisciplinary Biosciences Consortium (1903458). Research at UCL Great Ormond Street Institute of Child Health is supported by the NIHR Biomedical Research Centre at Great Ormond Street Hospital and UCL. The authors declare no competing financial interests.

# Additional information

## Funding

| Funder | Grant reference number | Author |
| --- | --- | --- |
| Medical Research Council | MR/P000843/1 | Tessa Crompton |
| Medical Research Council | MR/S037764/1 | Tessa Crompton |
| Biotechnology and Biological Sciences Research Council | BB/T020970/1 | Tessa Crompton |
| Biotechnology and Biological Sciences Research Council | 1903458 | Jasmine Rowell |

The funders had no role in study design, data collection and interpretation, or the decision to submit the work for publication.

## Author contributions

Jasmine Rowell, Conceptualization, Data curation, Formal analysis, Investigation, Visualization, Writing – original draft, Writing – review and editing; Ching-In Lau, Susan Ross, Diana C Yanez, Investigation; Oscar A Peña, Formal analysis; Benny Chain, Conceptualization, Supervision, Writing – review and editing; Tessa Crompton, Conceptualization, Supervision, Funding acquisition, Writing – original draft, Project administration, Writing – review and editing

## Author ORCIDs

Jasmine Rowell ⓘ https://orcid.org/0000-0001-7040-8528
Oscar A Peña ⓘ https://orcid.org/0000-0002-2582-0238
Benny Chain ⓘ https://orcid.org/0000-0002-7417-3970
Tessa Crompton ⓘ https://orcid.org/0000-0002-8973-4021

## Ethics

Animal experimentation was carried out under UK Home Office regulations (PPL PP7936049), following ethical review at UCL.

Reviewer #1 (Public review): https://doi.org/10.7554/eLife.93493.3.sa1
Reviewer #2 (Public review): https://doi.org/10.7554/eLife.93493.3.sa2
Author response https://doi.org/10.7554/eLife.93493.3.sa3

# Additional files

## Supplementary files

• Supplementary file 1. Antibody panel for staining thymus from 4 week mice.
• Supplementary file 2. Antibody panel for staining thymus from E18.5 mouse embryos.
• Supplementary file 3. Primers used for TCR sequencing protocol.
• Supplementary file 4. Reagents used for TCR sequencing protocol.
• MDAR checklist

## Data availability

TCR sequencing data are publicly available on UCL Research Data Repository at https://doi.org/10.5522/04/24161202.v1.

The following dataset was generated:

| Author(s) | Year | Dataset title | Dataset URL | Database and Identifier |
|---|---|---|---|---|
| Rowell J, Lau C, Ross S, Marcayata DY, Peña OA, Chain B, Crompton T | 2024 | Distinct T Cell Receptor (TCR) gene segment usage and MHC-restriction between foetal and adult thymus | https://doi.org/10.5522/04/24161202.v1 | UCL Research Data Repository, 10.5522/04/24161202.v1 |

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
