## [Editor Report · eLife Assessment]

This **important** manuscript provides an extensive and **convincing** analysis of the foetal and adult TCR repertoire in the mouse thymus. A potential implication of the work is that the earliest appearing T cells during ontogeny may have properties that are fundamentally distinct from those appearing later in life. The study will be of interest to immunologists concerned with T cell development and TCR repertoires.

---

## [Referee Report · Reviewer #1 (Public review)]

Summary:

The manuscript by Rowell et al aims to identify differences in TCR recombination and selection between foetal and adult thymus in mice. Authors sequenced the unpaired bulk TCR repertoire in foetal and adult mice thymi and studied both TCRB and TCRa characteristics in the double negative (DN, CD4-CD8-) and single positive (SP4 CD4+CD8- and SP8 CD4-CD8+) populations. They identified age-related differences in TCRa and TCRB segment usage, including a preferential bias toward 3'TRAV and 5' TRAJ rearrangements in foetal cells compared to adults who had a larger perveance for 5'TRAV segments. By depleting the thymocyte population in adult thymi using hydrocortisone, the authors demonstrated that the repertoire became more foetal like, they, therefore, argue that the preferential 5'TRAV rearrangements in adults may be resulting from prolonged/progressive TCRa rearrangements in the adult thymocytes. In line with previous studies, Authors demonstrate that the foetal TCR repertoire was less diverse, less evenly distributed and had fewer non-template insertions while containing more clonal expansions. In addition, the authors claim that changes in V-J usage and CDR1 and CDR2 in the DN vs SP repertoires indicated that positive selection of foetal thymocytes are less dependent on interactions with the MHC.

Strengths:

Overall, the manuscript provides an extensive analysis of the foetal and adult TCR repertoire in the thymus, resulting in new insights in T cell development in foetal and adult thymi.

Weaknesses:

Three major concerns arise:

(1) the authors have analysed TCR repertoires of only 4 foetal and 4 adult mice, considering the high spread the study may have been underpowered.

- The sample size was increased in the revised version

(2) Gating strategies are missing and

- These have now been provided in the revised version

(3) The manuscript is very technical and clearly aimed for a highly specialised audience with expertise in both thymocyte development and TCR analysis. Considering eLife is a scientific journal with a broader readership, Authors are recommended to provide schematics of the TCR rearrangements/their findings and include a summary of conclusions/implications of their findings at the end of each results section rather than waiting till the discussion. This will help the reader to interpret their findings while reading the results.

- These have now been included in the revised version

---

## [Referee Report · Reviewer #2 (Public review)]

Summary:

The authors comprehensively assess differences in the TCRB and TCRA repertoires in the fetal and adult mouse thymus by deep sequencing of sorted cell populations. For TCRB and TCRA they observed biased gene segment usage, less diversity, and greater repertoire sharing among individuals in fetal thymocytes. The TCRB repertoire was less evenly distributed and displayed more evidence of clonal expansions in fetal thymocytes. Both fetal and adult thymocytes demonstrated repertoire skewing in CD4 and CD8 as compared to DP thymocytes, which was attributed to MHC-I- vs MHC-II-restriction during positive selection. Effects of MHC-restriction were notably weaker in fetal thymocytes. The authors conclude that in multiple respects fetal repertoires are distinct from adult repertoires.

Strengths:

The analyses of the F18.5 and adult thymic repertoires are comprehensive with respect to the cell populations analyzed and the diversity of statistical approaches used to characterize the repertoires. Because repertoires were analyzed in pre- and post-selection thymocyte subsets, the data allowed assessment of repertoire selection at different developmental stages. Intriguing differences between fetus and adult are identified.

Weaknesses:

Some of the repertoire characteristics reported are already fairly well documented in the literature. Moreover, an unaddressed limitation of the study is that fetal thymocytes were analyzed at single time-point in their development. As a result, at least some of the conclusions about the fetal repertoire may be viewed not as general conclusions, but rather, due to the synchronous development of fetal thymocytes, as pertaining to the one day of fetal/early neonatal development assayed. Statements suggesting that (1) "progressive TCRa rearrangements occur less frequently in foetal DP cells" (Abstract), (2) "One possible explanation for this bias is that in the foetus progressive rounds of TCRa rearrangement are less common than in young adult" (Discussion), and (3) "Overall, the differences between the foetal and adult thymus TCR repertoires are consistent with the foetal thymus producing abT-cells ... with preference for particular gene segment usage" (Discussion), are oversimplified and potentially misleading.

---

## [Author Response]

The following is the authors’ response to the original reviews.

**Public Reviews:**

**Reviewer #1 (Public Review):**
Summary:The manuscript by Rowell et al aims to identify differences in TCR recombination and selection between foetal and adult thymus in mice. Authors sequenced the unpaired bulk TCR repertoire in foetal and adult mice thymi and studied both TCRB and TCRa characteristics in the double positive (DP, CD4+CD8+) and single positive (SP4 CD4+CD8CD3+ and SP8 CD4-CD8+CD3+) populations. They identified age-related differences in TCRa and TCRB segment usage, including a preferential bias toward 3'TRAV and 5' TRAJ rearrangements in foetal cells compared to adults who had a larger perveance for 5'TRAV segments. By depleting the thymocyte population in adult thymi using hydrocortisone, the authors demonstrated that the repertoire became more foetal like, they therefore argue that the preferential 5'TRAV rearrangements in adults may be resulting from prolonged/progressive TCRa rearrangements in the adult thymocytes. In line with previous studies, Authors demonstrate that the foetal TCR repertoire was less diverse, less evenly distributed and had fewer non-template insertions while containing more clonal expansions. In addition, the authors claim that changes in V-J usage and CDR1 and CDR2 in the DP vs SP repertoires indicated that positive selection of foetal thymocytes are less dependent on interactions with the MHC.Strengths:Overall, the manuscript provides an extensive analysis of the foetal and adult TCR repertoire in the thymus, resulting in new insights in T cell development in foetal and adult thymi.Weaknesses:Three major concerns arise:(1) the authors have analysed TCR repertoires of only 4 foetal and 4 adult mice, considering the high spread the study may have been underpowered.

Given the concerns of the reviewer we have sequenced more libraries and added more data to include repertoires from 7 embryos and 6 young adults (biological replicates from different sorts). We believe that including more replicates has indeed strengthened our study.

Our experimental approach was to sequence TCR transcripts, and in studies using RNA-sequencing of inbred mice, often only 3 individuals (biological replicates) are sequenced.

Our study sequenced from 7 foetal thymuses (generating TCRα and TCRβ repertoires from 4 FACS-sorted cell populations); 6 adult thymuses (generating TCRα and TCRβ repertoires from 4 FACS-sorted cell populations); and 5 adult thymuses from hydrocortisone-treated mice (generating TCRα and TCRβ repertoires from FACS-sorted CD3lo and CD3hi DP populations). We thus analysed 124 distinct repertoires from different populations and libraries, and many tens of thousands of unique sequences.

(2) Gating strategies are missing and

We have included gating strategies for cell-sorting as SFig7 and SFig8.

(3) the manuscript is very technical and clearly aimed for a highly specialised audience with expertise in both thymocyte development and TCR analysis. Authors are recommended to provide schematics of the TCR rearrangements/their findings and include a summary conclusions/implications of their findings at the end of each results section rather than waiting till the discussion. This will help the reader to interpret their findings while reading the results.

We have modified the manuscript to include a more general introductory paragraph (page 3) to introduce the reader to the topic and we have included brief summaries of the findings at the end of each result section (pages 7,9,10,12,13,15).

**Reviewer #2 (Public Review):**
Summary:The authors comprehensively assess differences in the TCRB and TCRA repertoires in the fetal and adult mouse thymus by deep sequencing of sorted cell populations. For TCRB andTCRA they observed biased gene segment usage and less diversity in fetal thymocytes. The TCRB repertoire was less evenly distributed and displayed more evidence of clonal expansions and repertoire sharing among individuals in fetal thymocytes. In both fetal and adult thymocytes they show skewing of V segment (CDR1-2) repertoires in CD4 and CD8 as compared to DP thymocytes, which they attribute to MHC-I vs MHC-II restriction during positive selection. However the authors assess these effects to be weaker in fetal thymocytes, suggesting weaker MHC-restriction. They conclude that in multiple respects fetal repertoires are distinct from and more innate-like than adult.Strengths:The analyses of the F18.5 and adult thymic repertoires are comprehensive with respect to the cell populations analyzed and the diversity of approaches used to characterize the repertoires. Because repertoires were analyzed in pre- and post-selection thymocyte subsets, the data offer the potential to assess repertoire selection at different developmental stages. The analysis of repertoire selection in fetal thymocytes may be unique.Weaknesses:(1) Problematic experimental design and some lack of familiarity with prior work have resulted in highly problematic interpretations of the data, particularly for TCRA repertoire development.The authors note fetal but not adult thymocytes to be biased towards usage of 3' V segments and 5'J segments. It should be noted that these basic observations were made 20 years ago using PCR approaches (Pasqual et al., J.Exp.Med. 196:1163 (2002)), and even earlier by others.

We have cited this manuscript (Introduction, page 5) which used PCR of genomic DNA to investigate some TCRα VJ rearrangements in foetal and adult thymus. In contrast, our study uses next generation sequencing of transcripts to investigate all possible combinations of TCRα and TCRβ VJ combinations in different sorted thymocyte populations ex vivo. The greater sensitivity of this more modern technology has thus enabled us to detect many more TCRαVJ rearrangements than the 2002 study, and to conclude on basis of stringent statistical testing that the foetal repertoire is enriched for 3’V to 5’J combinations (Fig. 4).

The authors also note that in fetal thymus this bias persists after positive selection, and it can be reproduced in adults during recovery from hydrocortisone treatment. The authors conclude that there are fewer rounds of sequential TCRA rearrangements in the fetal thymus, perhaps due to less time spent in the DP compartment in fetus versus adult. However, the repertoire difference noted by the authors does not require such an explanation. What the authors are analyzing in the fetus is the leading edge of a synchronous wave of TCRA rearrangements, whereas what they are analyzing in adults is the unsynchronized steady state distribution. It is certainly true, as has been shown previously, that the earliest TCRA rearrangements use 3' TRAV and 5'TRAJ segments. But analysis of adult thymocytes has shown that the progression from use of 3' TRAV and 5' TRAJ to use of 5' TRAV and 3' TRAJ takes several days (Carico et al., Cell Rep. 19:2157 (2017)). The same kinetics, imposed on fetal development, would put development of a more complete TCRA repertoire at or shortly after birth. In fact, Pasqual showed exactly this type of progression from F18 through D1 after birth, and could reproduce the progression by placing F16 thymic lobes in FTOC. It is not appropriate to compare a single snapshot of a synchronized process in early fetal thymocytes to the unsynchronized steady state situation in adults. In fact, the authors' own data support this contention, because when they synchronize adult thymocytes by using hydroxycortisone, they can replicate the fetal distribution. Along these lines, the fact that positive selection of fetal thymocytes using 3' TRAV and 5' TRAJ segments occurs within 2 days of thymocyte entry into the DP compartment does not mean that DP development in the fetus is intrinsically rapid and restricted to 2 days. It simply means that thymocytes bearing an early rearranging TCR can be positively selected shortly after TCR expression. The expectation would be that those DP thymocytes that had not undergone early positive selection using a 3' TRAV and a 5' TRAJ would remain longer in the DP compartment and continue the progression of TCRA rearrangements, with the potential for selection several days later using more 5'TRAV and 3'TRAJ.

We agree with this summary provided by the reviewer which corresponds closely to the points we made ourselves in the manuscript. Indeed, we discuss the synchronization and kinetics of first wave of T-cell development in Results page 13 and Discussion page 17, which was the rationale for the hydrocortisone experiment. We have also discussed findings from Carico et al 2017 in this context (see pages 13, 16, 17).

(2) The authors note 3' V and 5'J biases for TCRB in fetal thymocytes. The previously outlined concerns about interpreting TCRA repertoire development do not directly apply here. But it would be appropriate to note that by deep sequencing, Sethna (PNAS 114:2253 (2017)) identified skewed usage of some of the same TRBV gene segments in fetal versus adult. It should also be noted that Sethna did not detect significantly skewed usage of TRBJ segments. Regardless, one might question whether the skewed usage of TRBJ segments detected here should be characterized as relating to chromosomal location. There are two logical ways one can think about chromosomal location of TRBJ segments - one being TRBJ1 cluster vs TRBJ2 cluster, the other being 5' to 3' within each cluster. The variation reported here does not obviously fit either pattern. Is there a statistically significant difference in aggregate use of the two clusters? There is certainly no clear pattern of use 5' to 3' across each cluster.

We have included a statistical comparison of the aggregate TRBJ use between the J1 cluster and the J2 cluster (see SFig5) and Results page 9.

(3) The authors show that biases in TCRA and TCRB V and J gene usage between fetal and adult thymocytes are mostly conserved between pre- and post-selection thymocytes (Fig 2). In striking contrast, TCRA and TCRB combinatorial repertoires show strong biases preselection that are largely erased in post-selection thymocytes (Fig 3). This apparent discrepancy is not addressed, but interpretation is challenging.

I think the reviewer is referring to heatmaps for individual gene segment usage shown in Figure 2 in comparison to combinatorial usage shown in Figure 4. There is not a discrepancy in the data, but rather the differences between these two figures lie in the way in which the comparisons are made and visualised. The heatmaps in Figure 2A-D show mean proportional usage of each individual gene segment for each cell type in the two life stages, clustered by Euclidian distance. This visualisation clearly shows bias in foetal 3’ TRAV usage and 5’TRAJ usage (looking at areas of red, which have higher usage), with less pronounced enrichment for TRBV and TRBJ. The heatmaps also show differences in intensity between different cell populations in each life-stage.

In contrast, in Figure 4 the tiles show combinations with statistically significant (P<0.05) differences in mean counts for each VJ combination in each cell type between 7 foetal and 6 adult repertoires by Student’s t-test, after correcting for False discovery rate (FDR) due to multiple combinations. It is the case, that there are fewer significant differences in proportional combinatorial VxJ use between foetal and adult repertoires after selection. We find this an interesting finding and have expanded our discussion of this aspect of the data (page 10). More than half of the significant differences persist after repertoire selection, and the reduction in each individual SP population, of course in part reflects the lineage divergence.

(4) The observation that there is a higher proportion of nonproductive TCRB rearrangements in fetal thymus compared to adult is challenging to interpret, given that the results are based upon RNA sequencing so are unlikely to reflect the ratio in genomic DNA due to processes like NMD.

We have added two sentences to explain that transcripts of non-productive rearrangements are eliminated by nonsense-mediated decay (NMD), but some non-productive transcripts are detected in many studies of TCR repertoire sequencing, and we have cited three studies from different groups that document this (see Results, page 10-11). We have not commented on how the increase in non-productive TCR rearrangements in the foetal populations (in comparison to adult) relates to rearrangements in genomic DNA or NMD. We have likewise not commented on the possible significance or biological role of nonproductive TCR transcripts, but simply reported our findings.

(5) An intriguing and paradoxical finding is that fetal DP, CD4 and CD8 thymocytes all display greater sharing of TCRB CDR3 sequences among individuals than do adults (Fig 5DE), whereas DP and CD8 thymocytes are shown to display greater CDR3 amino acid triplet motif sharing in adults (with a similar trend in CD4).

As foetal DP, CD4SP and CD8SP TCRbeta repertoires have fewer non-template insertions and lower means CDR3 length, they are expected to share more CDR3 repertoires than their adult counterparts. However, in the case of CDR3 amino acid triplet motifs (k-mers) what is being analysed is the sharing of each possible individual k-mer. If k-mers are shared more in the adult for some populations, but CDR3 repertoires are shared more in the foetus, we think it means that some k-mers appear in many different CDR3 sequences in the adult, so that they are over-represented in multiple different CDR3s (presumably due to selection processes, although we agree that this is just an assumption).

The authors attribute high amino acid triplet sharing to the result of selection of recurrent motifs by contact with pMHC during positive selection. But this interpretation seems highly problematic because the difference between fetal and adult thymocytes is dramatic even in unfractionated DP thymocytes, the vast majority of which have not yet undergone positive selection. How then to explain the differences in CDR3 sharing visualized by the different approaches?

The TCRβ repertoire has been selected in the adult DP population through the process of β-selection, which is believed to involve immune synapse formation and MHC-interactions (Allam et al 2021,10.1083/jcb.201908108). We have now included this reference in the introduction to make this clear (page 4). However, we agree with the reviewer’s comments that it is challenging to explain the k-mer analysis and that we have not been able to actually show that increased k-mer sharing in the adult is a direct consequence of increased positive selection: it was our interpretation of this seemingly paradoxical finding. For clarity, we have therefore removed the k-mer analyses from the manuscript.

(6) The authors conclude that there is less MHC restriction in fetal thymocytes, based on measures of repertoire divergence from DP to CD4 and CD8 populations (Fig. 6). But the authors point to no evidence of this in analysis of TRBV usage, either by PC or heatmap analyses (A,B,D). The argument seems to rest on PC analysis of TRAV usage (Fig S6), despite the fact that dramatic differences in the SP4 and SP8 repertoires are readily apparent in the fetal thymocyte heatmaps. The data do not appear to be robust enough to provide strong support for the authors' conclusion.

We have written the text very carefully so as not to make the claim too strong, stating in the abstract: “In foetus we identified less influence of MHC-restriction on α-chain and β-chain combinatorial VxJ usage and CDR1xCDR2 (V region) usage in SP compared to adult, indicating weaker impact of MHC-restriction on the foetal TCR repertoire.” We are not saying that MHC-restriction does not impact VJ gene usage in foetal repertoires, but rather that it has less influence (particularly when compared to life-stage). Evidence for this comes from: [1] Heatmaps in Fig2A-D which show that all repertoires cluster first by life-stage ahead of cell type; [2] Fig3A and B: PCA of adult and foetal TCRβ VXJ combinations: All repertoires cluster by life-stage on PC1. PC2 separates adult repertoires by cell type (adult SP8 are positive on PC2 while adult SP4 are negative on PC2, and DP cells are between them) but for foetal repertoires the SP8 and SP4 are highly dispersed with some SP4 cells falling on positive side of PC2. Only foetal DP repertoires cluster tightly. [3] Fig6A-C: PCA of β−chain CDR1xCDR2 (corresponding to Vβ gene segment usage) again shows the same pattern. Adult repertoires separate by cell type on PC2, (SP8 positive on PC2, SP4 negative on PC2, with DP in between), but foetal SP8 repertoires are much more dispersed. [5] SFig6J-K: PCA of α−chain CDR1xCDR2 (Vα usage) frequency distributions: adult repertoires cluster together and are separated by cell type on PC2 (SP4 positive, SP8 negative), but foetal populations are highly dispersed and fail to cluster by cell type on either axis. [6] We have additionally added new PCA analyses to explore differences in MHC-restriction between foetal and adult SP populations. This is shown in the new Figure 7. We reasoned that in a PCA that included foetal and adult repertoires together, the foetal repertoires might not segregate by SP cell type (MHC-restriction) because of their overall bias towards particular VJ combinations, which would mean that effectively the PCA would be imposing adult MHC restriction on the foetal repertoires. We therefore carried out PCA in which we analysed the adult repertoires separately from the foetal repertoires. As expected for adult repertoires, PCA separated SP4 repertoires from SP8 repertoires on PC1 in each comparison (β-chain VxJ (Fig. 7B), α-chain VxJ (Fig. 7F), β-chain CDR1xCDR2 (V region) (Fig. 7H) and α-chain CDR1xCDR2 (V region) (Fig. 7L)). In contrast, for foetal TCRα repertoires (α-chain VxJ and α-chain CDR1xCDR2 (V region)), PCA failed to separate SP4 from SP8 repertoires on PC1 or PC2, so we did not detect impact of MHC-restriction on foetal TCRβ repertoires (Fig. 7E and K). For foetal TCRβ repertoires, PCA separated SP4 β-chain VxJ from SP8 on PC2, accounting for only 11.1% of variance (Fig. 7A) (in contrast to the 44.2% of variance accounted for by MHC-restriction in adult β-chain VxJ PCA (Fig. 7B)). Thus, in adult repertoires ~4-fold more of the variance in β-chain VxJ usage can be accounted for by MHC-restriction than in foetal repertoires. PCA of foetal β-chain CDR1xCDR2 (V region) separated SP4 from SP8 on PC1, accounting for 28.8% of variance, whereas in PCA of adult β-chain CDR1xCDR2, MHCrestriction accounted for 56.1% (>2-foldmore than in foetus). Thus, even when we considered only V-region usage alone, we detected a stronger influence of MHC-restriction on the TCRβ repertoire in adult compared to foetal thymus.

**Reviewer #3 (Public Review):**
Summary:This study provides a comparison of TCR gene segment usage between foetal and adult thymus.Strengths:Interesting computational analyses was performed to find interesting differences in TCR gene usage within unpaired TCRa and TCRb chains between foetal and adult thymus.Weaknesses:This study was significantly lacking insight and interpretation into what the data analysed actually means for the biology. The dataset discussed in the paper is from only two experiments. One comparing foetal and adult thymi from 4 mice per group and another which involved hydrocortisone treatment. The paper uses TCR sequencing methodology that sequences each TCR alpha and beta chains in an unpaired way, meaning that the true identity of the TCR heterodimer is lost. This also has the added problem of overestimating clonality, and underestimating diversity.

We have discussed the limitations and benefits of our approach of sequencing TCRβ and TCRα repertoires separately in the Discussion (page 19). This approach allows the analysis of thousands of sequences from different cell types and different individuals at relatively low cost. We have made no claims in our manuscript about overall diversity or pairing, and given that each chain’s gene locus rearranges at a different time point in development, we believe it is of interest to consider the repertoires individually within this context.

Limited detail in the methods sections also limits the ability for readers to properly interpret the dataset. What sex of mice were used? Are there any sex differences? What were the animal ethics approvals for the study?

We have included this information in the Methods (page 19). Both sexes were used and we found no sex differences, although that was not the focus of our study. All animal experimentation in the UK is carried out under UK Home Office Regulations (following ethical review). This is included in the Methods (page 19).

**Recommendations for the authors:**

**Reviewer #1 (Recommendations For The Authors):**
Major points:- Group sizes are very small (4 foetal and 4 adult mice). Considering the spread in TCR analysis (eg fig 1 B-H, Sup figures 2-4), the study is likely underpowered as it often looks like one mouse prevents or supports a statistical difference. Authors should therefore consider increasing the group size.

We have sequenced more libraries and included more data, from 7 foetal and 6 young adult animals (biological replicates).

- The authors should include a gating strategy for their sorted cells. This is essential to verify the quality of their findings.

We have added this to the Methods and SFig7 and SFig8.

Authors should include a summary sentence at the end of each result section which interprets the main finding. Furthermore, the manuscript would greatly benefit from a schematic figure of their main findings, particularly with regards to the rearrangements and selection differences in foetal and adult thymi.

We have added a summary sentence to the end of each results section.

- Authors should be more careful with their claim that MHC has less of an effect foetal TCR selection. Authors demonstrated that there is a difference in VJ recombination between the foetal and adult TCR repertoire, skewing the foetal TCR repertoire to certain variable and junctional segments. Since both CDR1 and CDR2 are encoded by the variable gene, this is likely to affect their ability to interact with the MHC during positive selection. Have Authors considered whether the selection process is actually a bystander effect of the differences in the rearrangement process? One way to support the authors claim is to demonstrate that mice with an alternative MHC background, have similar foetal/adult gene rearrangements but a different TCR repertoire in the SP populations.

Time and resources have prevented us from repeating our experiments in another strain of inbred mice. However, we note that a previous PCR study that showed 3’TRAV to 5’TRAJ bias in foetal repertoires was carried out in BALB/c mice (Pasqual JEM 2002). We have added this point to the Discussion (page 17).

- (supplementary) tables have not been provided.

Supplementary Tables were uploaded with the submission. STables 1 and 2 show antibodies used for cell sorts and STable 3 primers used.

Moderate points:- The loading plots in Figure 3 onward are visually strong. Authors could consider including an V and J (separate) loading plots for Figure 3 E, F and G to demonstrate preferential V and J usage.

We have included additional loading plots in Figure 7 for the new PCA we have added (see Fig. 7C, D,I and J).

- "the proportion of non-productive rearrangements was higher in the foetal SP8 population than adults (Fig 5A)" Authors should explain how non-productive TCRs end up in SP populations as they need to pass positive and negative selection which both require interactions between the TCR and the MHC.

As we used RNA sequencing in our study, we did not comment on how the increase in nonproductive TCRbeta rearrangements in the foetal populations (in comparison to adult) relates to rearrangements in genomic DNA or to nonsense-mediated decay (NMD) that is believed to down-regulate transcripts of non-productively rearranged TCR. We have not commented on the possible significance or biological role of non-productive TCR transcripts, but simply reported our findings.

- Authors have studied CDR3 sequential amino acid triplets (k-mers). However, CDR3 regions are longer than 3 amino acids in length, hence authors should provide (1) an overview/comparison of the identified k-mers in foetal or adult thymocytes (2) explain how different k-mers relate to each other, eg whether they are expressed in the same TCR. Have authors considered using alternative programs to identify CDR3 motifs that are based on the full CDR3amino acid sequence, eg TCRdist provides motifs and indicated which amino acids are germline encoded or inserted.

In light of this comment from this reviewer and also comments from Reviewer 2, we have removed the comparison of k-mers from the manuscript. Please see response to point 5 of Reviewer 2.

- The term "innate-like" is confusing as it implies that foetal cells are not antigen specific.However, once in the circulation, foetal cells will respond in an antigen-specific manner.Hence authors should use another term.

We have removed the term “innate-like” from the abstract and the first time we used it in the first paragraph of the Discussion. However, the second time we used the term, we are actually taking it from the manuscript we cited (Beaudin et al 2016) and in this case we left it in. We agree that foetal cells are likely to respond in an antigen-specific manner.

- To support their hypothesis in the discussion "However, as TCRd gene segments are nested.... so that 5' TRAV segments are not favoured" can authors confirm that there are indeed less yd T cells in the foetal repertoire?

We have removed this section from the discussion, because although it is interesting, it is highly speculative, and the manuscript is already quite complicated to interpret.

Minor points:- The authors may find the publication by De Greef 2021 PNAS of interest to identify TRBD segments- Authors need to clarify that they mean CDR3-beta in the sentence "The mean predicted CDR3 length.... compared to young adult"

We have included new data in the manuscript to show that mean CDR3 length is lower in all foetal populations of beta (Fig5C) and alpha (SFig5C) and clarified which we are referring to in the text.

- Authors should bring the section "During TCRb gene rearrangement, these segments.... Initiating the sequence of rearrangements" forward and include a schematic." Forward to figure 2 and provide the reader with a visual schematic of the foetal vs adult recombination events.- Discussion: "The first wave of foetal abT-cells that leave the thymus... tolerant to both self and maternal MHC/antigens". Have Authors considered the alternative hypothesis published by Thomas 2019 in Curr Opin System Biol that the observed bias could potentially provide better protection against childhood pathogens?

We have indeed considered this, as stated in the first paragraph of the Discussion “The first wave of foetal αβT-cells that leave the thymus must provide early protection against infection in the neonatal animal”. We have now cited the Thomas 2019 study.

- Discussion: Authors should rephrase the sentence "The transition from DP to SP cell in the foetus.... From DN3 to SP cell may be slower" as it is unclear what the authors mean.

We have rephrased this (see page 17)

- Discussion "TRAV and TRAJ Array" do authors mean "TRAV and TRAJ area"?

We did indeed mean array (as in series of gene segments) but we have changed the wording for clarity (page 14).

- Methods, Fluorescence activated cell sorting: can authors clarify whether they stained, sorted and sequenced the full thymus and /or specify how many cells were included. Can authors also explain why foetal and adult cells were treated differently (eg the volume of master mix)?- Methods Fluorescence activated cell sorting authors should specify what they mean with "mastermix of either 1:50 (foetal thymus) or 1:100 (adult thymus)". Does this mean all antibodies in the foetal mastermix were 1:50 and all antibodies in the adult master mix were 1:100? If so, why were different concentrations used and why were antibodies not individually titrated before use?

We have clarified the methods and antibodies used are listed with clones in supplementary tables.

Figures:- Several figures did not fit on the page and therefore missed the top or side- Figure 1A: missing a label on the Y axis

This is visible

- Figure 2A-D: please indicate the 5' and 3' terminus in each graph. The cell type legend should include two separate colours for the two DP populations.

We have added 5’ and 3’ labels. The two DP populations are clearly labelled.

- Figure 4: please indicate the 5' and 3' terminus in each graph.

We have added 5’ and 3’ labels.

- Figure 5C: y axis should read mean CDR3B length (aa), Figure 5D and E: y axis should read Jaccard Index CDR3B, Figure 5 F and G: y axis should read Jaccard index CDR3B k-mers. Same comment for Sup Fig 5 but then CDR3a.

We have added these labels for both Figure 5 and Supplementary Figure 6 (was SFig5 previously).

- Figure 6C top label should read CDR1B x CDR2B with highest contribution

We have added this label.

- Figure 7: please indicate the 5' and 3' terminus in each graph.

We have added 5’ and 3’ labels. This is now Figure 8, as we have added new analyses (new Figure 7).

- Supplementary Figure 1-4 are missing a colour legend next to the graphs.

We have added the legends in.

**Reviewer #2 (Recommendations For The Authors):**
(1) The authors need to provide better support for the notion that the fetal thymus produces ab T cells with properties and functions that are distinct from adult T cells. There are several ways they might provide a more meaningful assessment: (1) They could analyze the fetal repertoire at multiple time points. (2) They could compare instead the steady state distributions in early postnatal and adult thymus samples. (3) They could compare the peripheral T cell repertoires in the first week of life versus adult. This last approach would allow them to draw the most impactful conclusion.

We appreciate these suggestions. Sadly, it is beyond our budget for the current manuscript and beyond the scope of our current study that we believe provides interesting new information.

(2) Fig S2D shows TRBJ1-4 in black lettering meant to indicate no significant difference whereas the figure shows use of this gene segment to be elevated in adult. I believe TRBJ1-4 should be in blue lettering.

This is now coloured correctly.

(3) The figure call out on p11 (Fig5I-J) should be H-I.

This is now corrected.

(4) Please indicate in the main text that Jaccard analysis in Fig 5 D-E is for TCRB.

This is now corrected.

(5) The analysis of usage of TCRB CDR1xCDR2 combinations in Fig6D is said to "reflect the bias observed in their TRBV gene usage (Fig 2C)". Isn't it the case that every TRBV gene presents a distinct CDR1xCDR2 combination, meaning that there is no difference between TRBV usage and TRBV CDR1xCDR2 usage? If so, please make this clearer.

Yes, this is the case, we have made this clearer in the text.

**Reviewer #3 (Recommendations For The Authors):**
In general, although there is lots of interesting analyses that can be done with these large datasets, I feel as though the authors did not fully interpret the real meaning and significance of many of these results. Whilst there were some speculation on why a foetal repertoire might be different to those of adults in the discussion sections, the rationale for each individual analyses was not clearly explained. I would suggest that the rationale and a thorough explanation of each analyses be added to the results section, including a finishing sentence on what it means.

We have added short summaries to each results section to make the points we are making clearer.

The authors did not mention how many cells were sorted for from each thymus for sequencing. Was the cell number normalised between each population? As this might have an influence on various downstream measurements of diversity, evenness and clonality, if there is a sampling issue.

This is explained in the methods. We used sampling to allow comparisons between repertoires of different sizes, and this is also explained in the methods.

The authors should include the cell sorting profiles and example flow cytometry plots, including gating strategies and the post sort purity of each sorted population.

We have included sorting strategies in the methods (SFig7 and SFig8).

I think the manuscript could also be improved if there were some basic characterisation of foetal vs. adult thymus development. How many thymocytes are in a foetal vs adult thymus at the timepoints chosen?I think there were some interesting findings in this paper. Given that overall, the foetal thymus appeared to be less diverse than that of the adult, one question I thought would be interesting to discuss was the overlap between the two repertoires. Is the foetal thymus simply a sub-fraction of the adult repertoire or is it totally distinct with no overlapping sequences?

Our analyses indicate that the repertoires are actually different. This is evident in Fig4 and in PCA loading plots shown in Fig, 3C and new Fig. 7C, D, I and J.

I think that some of the interpretation in the results section may be a bit vague. "When we compaired by thymocyte population, each adult population clustered together, with adult SP4 separating from adult SP8 on PC2 and DP cells scoring in between, suggesting that PC2 might correspond to MHC restriction of the adult populations." - whilst I think I know what the authors mean, I do believe that this could be explained in clearer detail and more explicit. SP4 and SP8 are known to be positively selected in the thymus on distinct MHC class I and MHC class II molecules for example.

We have tried to clarify the text describing that PCA and additionally added a new Figure (new Fig. &) to compare the influence of MHC-restriction on the TCR repertoire in foetal and adult thymus.

In the methods section, the age and sex of mice used were not explained at all. What was used in the experiment? Are there any sex differences?

Age and sex of mice is given in the methods. We have not detected sex differences.

This is a huge omission from the manuscript. In general, I don't believe the methods section has described the analysis in sufficient detail for replication. All analysis code and data should be publicly accessible and be in a format that allows for the reader to replicate the figures in the paper upon running the code. Perhaps even allowing them to run their own TCR datasets. Overall, I think the manuscript needs some rewriting to include additional details and deeper interpretation of each individual analyses.

Sequencing data files will be made publicly available on UCL Research Data Repository.